# Enhancing Communication Compression via Discrepancy-aware Calibration for Federated Learning

**Zhiyi Wan**[*,1]**, Yijia Chi**[*,1]**, Liang Li**[2]**, Wanrou Du**[1]**, Miao Pan**[3]**, Xiaoqi Qin**[1]
[1]Beijing University of Posts and Telecommunications
[2]Peng Cheng Laboratory
[3]University of Houston

## Abstract

Federated Learning (FL) offers a privacy-preserving paradigm for distributed model training by enabling clients to collaboratively learn a shared model without exchanging their raw data. However, the communication overhead associated with exchanging model updates remains a critical challenge, particularly for devices with limited bandwidth and battery resources. Existing communication compression methods largely rely on selection rules based on magnitude or randomness. For example, Top-k drops the elements with small magnitude, while low-rank methods such as ATOMO and PowerSGD truncate singular values with small magnitude. However, these rules do not account for the discrepancy between the compressed and the original outputs, which can lead to the loss of important information. To address this issue, we propose a novel discrepancy-aware communication compression method that enhances performance under severely constrained communication conditions. Each client uses a small subset of its local data as calibration data to directly measure the output discrepancy induced by dropping candidate compression units and uses it as a compression metric to guide the selection. By integrating this strategy, we can enhance existing mainstream compression schemes, enabling more efficient communication. Empirical results across multiple datasets and models show that our method achieves a significant improvement in accuracy under stringent communication constraints, notably an $18.9\%$ relative accuracy improvement at a compression ratio of $0.1$, validating its efficacy for scalable and communication-efficient FL. Our code is available at https://github.com/wzy1026wzy/Discrepancy-aware-Compression-for-FL.

## 1 Introduction

Federated Learning (FL) has emerged as a paradigm for distributed learning, enabling models to be trained across decentralized devices without requiring data to leave the local device. This decentralized approach reduces data privacy concerns by transmitting parameter updates instead of raw data during training. However, FL presents unique challenges due to the limited communication capacity of client devices, which can result in a high communication burden during the training process. In this context, a key challenge in FL is optimizing the trade-off between model accuracy and communication efficiency. Consequently, most FL studies have reduced server–client synchronization frequency since the inception of FL (McMahan et al., 2017). Clients often train for multiple steps or epochs locally before transmitting and aggregating their updates.

One of the key strategies to alleviate the communication burden in FL is the use of communication compression techniques, many of which are adapted from distributed learning's communication optimization methods. These methods, such as Top-k (Lin et al., 2018; Alistarh et al., 2018; Renggli et al., 2019), Random-k, Random-Block, ATOMO (Wang et al., 2018), and PowerSGD (Vogels et al., 2019), aim to reduce the volume of data exchanged between clients and the server in FL. Top-k, Random-k, and Random Block are element-wise sparsification methods that focus on selecting

---

*  These authors contribute equally to this work.

elements to drop based on magnitude or random selection. ATOMO and PowerSGD are low-rank methods that transmit the low-rank decomposition of the updates, retaining only the singular values with larger magnitudes and their corresponding singular vectors.

However, all of these communication compression methods in FL follow selection rules based on magnitude or randomness, to determine which compression units to retain for further upload and aggregation. In the case of element-wise sparsification methods like Top-k, the compression units correspond to individual elements. And for low-rank methods like ATOMO, the compression units consist of singular triplets, each of which includes a singular value and its corresponding singular vectors. While these selection rules are simple and efficient, they ignore the impact of dropping specific compression units on the output discrepancy. As a result, important units may be discarded, leading to significant compression losses and harming the model's performance. This observation is consistent with the well-known gap between Top-$k$ and Random-$k$ in practical behavior, even though they share the almost same worst-case bounds and convergence guarantees (Beznosikov et al., 2023; Condat et al., 2022; Li & Li, 2023; Qian et al., 2021; Yi et al., 2025; Karimireddy et al., 2019). This is because the coordinates with the largest magnitudes always captures more informative updates. Our method pushes this intuition further, as detailed in Section 3.2.

In FL, there have been many studies that tailor compression methods to the server–client training loop. For example, FedFQ (Li et al., 2024) introduces parameter-level adaptive quantization, selecting per-parameter bit widths via constraint-guided simulated annealing; FedAQ (Qu et al., 2024) jointly optimizes uplink and downlink bit widths under energy constraints, derives convergence guarantees, and adopts a schedule with decreasing uplink and increasing downlink precision across rounds; FedMPQ (Chen & Vikalo, 2024) brings mixed-precision quantization to heterogeneous FL, assigning layer-wise bit widths while the server allocates per-client bit budgets across rounds; AdapComFL (Zhuansun et al., 2024) employs compression based on bandwidth prediction; and HGC (Hu et al., 2024) unifies sparsification, quantization, and entropy coding within a hybrid framework. These approaches address several meaningful challenges in FL. However, they still rely on selection rules based on magnitude or randomness, without leveraging FL-specific characteristics such as low-frequency communication to address the limitations of existing compression methods.

This limitation partly stems from not fully leveraging a key characteristic that distinguishes FL from traditional distributed learning—its infrequent communication. Unlike data-center training, FL allows clients to perform multiple local updates before a global aggregation. There is a fundamental difference in the type of communicated information: distributed learning typically transmits gradients, while FL typically transmits parameter updates accumulated over multiple local steps (McMahan et al., 2017; Li et al., 2020; Dandi et al., 2022). As a result, the content to be compressed in FL carries little direct loss-related information, further reducing the correlation between magnitude and importance as discussed in A.2. Consequently, employing selection rules that rely on magnitude or randomness to determine which compression units to drop is fundamentally inadequate. Furthermore, the infrequent communication in FL makes additional local computation a practical way to maximize the value of each transmitted bit, which would be too costly in distributed learning.

Building on this observation, we propose a novel discrepancy-aware compression strategy for communication-efficient FL that enhances performance under severely constrained communication conditions. Each client uses a small subset of its own local data as a calibration dataset to directly measure the output discrepancy of dropping different compression units, guiding the compression selection process. This discrepancy-aware selection replaces selection rules based on magnitude or randomness with decisions grounded in output sensitivity, leading to more efficient communication and improved model accuracy under constrained communication budgets.

Our key contributions are as follows:

- We propose a novel discrepancy-aware compression strategy for communication-efficient FL. Each client uses a small local calibration dataset to directly measure and minimize the impact of compression on output discrepancy. This approach addresses the drawback of existing FL compression methods that use magnitude as the importance evaluation.

- Our proposed strategy can seamlessly enhance existing compression schemes as a plug-in module. We demonstrate its versatility by improving two representative methods: the sparsification method Top-k and the low-rank decomposition method ATOMO.

- We conduct extensive experiments on diverse models and datasets. The results show that our approach achieves superior accuracy under stringent communication budgets. For instance, on CIFAR-10 with the ViT-tiny model, it yields an $18.9\%$ relative accuracy improvement at a compression ratio of $0.1$ compared to the baseline.

## 2 RELATED WORK

**Federated Learning (FL).** A classic way to reduce the communication frequency in distributed learning is to allow clients to perform multiple local Stochastic Gradient Descent (SGD) steps before synchronization. The seminal work on FL was introduced by Konečný et al. (2016), who presented a novel paradigm for distributed optimization in machine learning. Building on this framework, FedAvg (McMahan et al., 2017) introduced the idea of reducing communication overhead by allowing clients to perform multiple local iterations between synchronizations, maintaining accuracy while improving wall-clock efficiency. Subsequent theory on local/periodic averaging sharpened this picture: local SGD can match mini-batch SGD in gradient complexity and dramatically reduce communication rounds (Stich, 2019), with tighter nonconvex analyses and adaptive schedules in periodic averaging (Lin et al., 2020). Recent works, including AdaComm (Wang & Joshi, 2019), FedPAQ (Reisizadeh et al., 2020), FedProx (Li et al., 2020), SCAFFOLD (Karimireddy et al., 2020), FedBuff (Nguyen et al., 2022), and CTUS (Wang et al., 2024), use drift correction, buffering, and adaptive strategies to improve performance under partial participation and heterogeneous clients. Overall, communication-delay methods aim to reduce uplink and downlink traffic. Our method aligns with the general trend in FL of reducing communication cost and improving the trade-off between communication frequency and model accuracy.

**Communication-efficient FL.** Limited uplink bandwidth and device energy constraints make communication a critical bottleneck in FL. To mitigate this, most approaches focus on traditional communication optimization techniques, such as quantization and compression methods. Quantization methods aim to reduce the encoding bit-width of parameter updates. Representative works include QSGD (Alistarh et al., 2017) and SignSGD (Bernstein et al., 2018), while recent FL-specific approaches (Li et al., 2024; Cao et al., 2024; 2025; Qu et al., 2024; Chen & Vikalo, 2024) perform fine-grained adaptive quantization across parameters to optimize communication efficiency. Compression methods, including sparsification and low-rank decomposition, have also been widely adopted in FL, originally developed for distributed training. These methods perform compression operations on the matrix itself and its transformed form, respectively. Sparsification methods reduce communication by transmitting only a small subset of updates. The most common approaches are Random-k and magnitude-driven Top-k (Lin et al., 2018; Alistarh et al., 2018; Renggli et al., 2019). Beyond these general methods, FL-tailored variants such as STC (Sattler et al., 2019), FedSPA (Hu et al., 2021a), AdapComFL (Zhuansun et al., 2024), and HGC (Hu et al., 2024) have explored adaptive compression ratio, bidirectional compression, secure aggregation, and differential privacy mechanisms. Low-rank methods exploit the low-rank property of weight update matrices, reducing communication load proportional to the rank. ATOMO (Wang et al., 2018) was the first to propose this sparsification framework, and PowerSGD (Vogels et al., 2019) quickly approximates low rank using power iteration, widely applied in industrial and open-source distributed training. However, these compression schemes generally rely on selection rules based on based on magnitude or randomness, often resulting in higher compression loss.

## 3 DISCREPANCY-AWARE COMPRESSION SCHEMES FOR FL

In this section, we first present the objectives and motivation behind our approach. We then introduce a compression method that estimates the induced output discrepancy from dropping candidate compression units—either an element or a singular triplet—on a small local calibration dataset, and selectively drops candidates. Subsequently, we derive a compression metric tailored for different network architectures and compression techniques. Building on this framework, we enhance commonly used element-wise sparsification and low-rank compression methods, while ensuring compatibility with efficient low-rank approximations such as PowerSGD.

### 3.1 OBJECTIVE: MINIMIZE THE OUTPUT DISCREPANCY

For a Transformer layer with a matrix $W_0$, a parameter update matrix $W$ and the corresponding input activations $X$, the output will be $Y = (W_0 + W)X$. For a CNN with kernel size $F$, stride $s$, and padding size $p$, we model the layer as cross-correlation. Let $W_{k,c,0}, W_{k,c} \in \mathbb{R}^{F \times F}$ denote the original kernel and its update from input channel $c$ to output channel $k$ and $\bar{X}$ be $X$ zero-padded by $p$ pixels on each border. Then the output at channel $k$ is $Y_k[u, v] = \sum_{c=1}^{C_{\text{in}}} \sum_{i=0}^{F-1} \sum_{j=0}^{F-1} (W_{k,c,0} + W_{k,c})[i,j] \, \bar{X}_c[us + i, \ vs + j]$. Given a compressed update $\widehat{W}$ producing $Y'$, our goal in both cases is to minimize the output discrepancy on a calibration set, measured by the Frobenius norm:

$$\min_{\widehat{W}} \ \mathcal{L}_{\text{comp}}(W - \widehat{W}) \ = \ \sum_X \left\| \Delta Y \right\|_F^2 \ = \ \sum_X \left\| Y - Y' \right\|_F^2.$$

### 3.2 MOTIVATION: LOW MAGNITUDE $\neq$ LOW IMPORTANCE

Existing compression methods decide which compression units to drop using selection rules that do not incorporate loss- or discrepancy-related information. For example, Top-k drops the small-magnitude elements, while low-rank methods such as ATOMO and PowerSGD truncate small-magnitude singular values. Without considering the output discrepancy induced by compression, these rules can drop compression units whose removal causes a large change in the layer output, leading to inefficient utilization of communication resources.

In FL, this challenge is exacerbated by a fundamental difference from traditional distributed learning (DL), where communication is frequent and gradients directly dictate the importance of each coordinate. In contrast, FL uses parameter updates across multiple local steps, resulting in compressed content that does not directly correlate with the gradient magnitudes, as detailed in A.2. Therefore, magnitude-based selection rules, such as those employed in the Top-k and other methods, fail to appropriately capture the true importance of the components being compressed, further limiting the accuracy and communication efficiency trade-off in FL.

**A Motivating Example of Element-wise Sparsification.** Selecting the largest elements of a matrix $W \in \mathbb{R}^{m \times d}$ does not guarantee minimal distortion, because the actual impact on outputs $Y = WX$ depends on the input activations $X \in \mathbb{R}^{d \times n}$. Write $X$ by rows as $X = [f_1^\top, \ldots, f_d^\top]^\top$. The induced output discrepancy in the form of Frobenius loss from removing an element $w_{i,j}$ equals

$$\mathcal{L}_{\text{comp}}(w_{i,j}) \ = \ \| (W - W_{\setminus w_{i,j}})X \|_F^2 \ = \ \| w_{i,j} f_j \|_F^2 \ = \ w_{i,j}^2 \, \| f_j \|_F^2.$$

Consider two candidate elements $w_{i1} = 0.1$ and $w_{i2} = 10$ with their corresponding input activation feature $\| f_1 \|_F = 10^3$ and $\| f_2 \|_F = 10^{-3}$. If we drop $w_{i1}$, the induced output discrepancy would be calculated as $\mathcal{L}_{\text{comp}}(w_{i1}) = (0.1 \cdot 10^3)^2 = 10^4$. On the other hand, if we drop $w_{i2}$, the output discrepancy would be $\mathcal{L}_{\text{comp}}(w_{i2}) = (10 \cdot 10^{-3})^2 = 10^{-4}$. The magnitude-based Top-k rule would keep $w_{i2}$ and drop $w_{i1}$. However, this choice induces an output discrepancy $10^8$ times larger than the alternative, revealing a clear gap between selecting by element magnitude and selecting by output discrepancy induced. The argument holds whether the Top-k is applied per row or globally over all entries of $W$. Random-k can fare worse by discarding high-impact entries purely by chance.

**A Motivating Example of Low-rank Decomposition.** Truncating smaller singular values is not necessarily safe if the associated singular vectors align with high-energy input directions. Let the SVD of $W \in \mathbb{R}^{m \times d}$ be $W = \sum_{i=1}^{2} \sigma_i u_i v_i^\top$. A standard low-rank truncation would remove the smaller singular value $\sigma_1$. However, the induced output discrepancy in the form of Frobenius loss from dropping component $i$ equals

$$\mathcal{L}_{\text{comp}}(\sigma_i) \ = \ \| (W - W_{\setminus \sigma_i})X \|_F^2 \ = \ \| \sigma_i \, u_i v_i^\top X \|_F^2 \ = \ \sigma_i^2 \, \| u_i (v_i^\top X) \|_F^2 \ = \ \sigma_i^2 \, \| v_i^\top X \|_F^2.$$

Consider two candidate singular values $\sigma_1 = 1$ with $\| v_1^\top X \|_F = 10^3$ and $\sigma_2 = 100$ with $\| v_2^\top X \|_F = 10^{-3}$. If we drop $\sigma_1$, the induced output discrepancy would be calculated as $\mathcal{L}_{\text{comp}}(\sigma_1) = 1 \cdot 10^3 = 10^3$. On the other hand, if we drop $\sigma_2$, the output discrepancy would be $\mathcal{L}_{\text{comp}}(\sigma_2) = 100 \cdot 10^{-3} = 10^{-1}$. Therefore, removing the smaller singular value incurs an output discrepancy $10^8$ times larger than removing the larger one.

---

**Algorithm 1:** Communication-efficient FL with discrepancy-aware compression

---

**Input** : Number of communication rounds $T$; Client set $\mathcal{C}$; Global model $W^0$; Learning rate $\eta$;
Local training epochs $E$; Communication budget B;
**Output:** Final global model $W^T$.

---

1 **for** $t = 1, \ldots, T$ **do**
2      Server samples clients $\mathcal{C}_t \subseteq \mathcal{C}$ and broadcasts global model $W^{t-1}$.
3      **for** *each Client* $k \in \mathcal{C}_t$ **in parallel do**
4          Local training based on $W^{t-1}$ using private training data for $E$ epochs, obtain $\Delta W_k$.
5          Discrepancy-aware compression based on local calibration dataset, obtain $\widehat{\Delta W}_k$.
6          Transmit $\widehat{\Delta W}_k$ to Server.
7      Server aggregates clients' updates $W^t \leftarrow W^{t-1} + \sum_{k \in \mathcal{C}_t} \frac{n_k}{m_t} \widehat{\Delta W}_k, m_t = \sum_{k \in \mathcal{C}_t} n_k,$
     where $n_k$ is the number of samples on client $k$ in round $t$.

---

These motivating examples indicate that existing compression strategies remove compression units without regard to their actual contribution to the output, leading to disproportionate output discrepancy for a given communication budget. To address this limitation, we adopt a calibration-based perspective that explicitly links the selection of compression units to the output discrepancy. This perspective unifies disparate compressors and highlights why selection rules based solely on magnitudes or randomness are fundamentally insufficient.

### 3.3 KEY DESIGN

As outlined in Algorithm 1 and 2, we seek a unified, compression-aware principle for communication compression that directly scores each candidate compression unit by the output discrepancy it would induce on a small calibration dataset. Let $W$ be a layer update to be compressed, and $X$ denote the calibration activations. We recommend that each client randomly select a small subset of its local training dataset in each round to compute calibration activations, which are then used for scoring compression units. This choice is empirically evaluated in Section 4.4. For each candidate compression unit $u$ (whether an element or a singular triplet), we use the output discrepancy on a calibration set as a compression metric to estimate its removal cost:

$$\mathcal{L}_{\text{comp}}(u) \;=\; \sum_{\text{cal } X} \left\| \Delta Y \right\|_F^2 \;=\; \sum_{\text{cal } X} \left\| Y - Y' \right\|_F^2.$$

Then we rank these units by their compression metric $\mathcal{L}_{\text{comp}}(u)$, and select the subset that approximately minimizes $\mathcal{L}_{\text{comp}}(u)$ under the given budget. This leads to a simple drop rule: keep units with large $\mathcal{L}_{\text{comp}}(u)$, and drop those with small $\mathcal{L}_{\text{comp}}(u)$. Below we derive $\mathcal{L}_{\text{comp}}(u)$ for different granularities that encompass common compressors. Notably, like most compression methods, our approach can operate under either a global or layer communication budget.

In addition, Our method is fully compatible with classic error feedback scheme. Concretely, each client maintains a residual vector that accumulates the compression error from previous rounds. At each communication round, each client first forms a compensated update by adding the residual to its current local update. The compensated update is then passed through the proposed compression operator, yielding the transmitted message. And the residual is updated by subtracting the transmitted compressed update from the compensated update.

### 3.4 COMPRESSION METRIC

**For Element-wise Sparsification in Transformer Layers.** Dropping a single element $w_{i,j}$ replaces $W$ by $W_{\backslash w_{i,j}} = W - w_{i,j} e_i e_j^\top$. Then

$$\Delta Y_{w_{i,j}} = (W - W_{\backslash w_{i,j}})X \;=\; w_{i,j}\, e_i e_j^\top X \;=\; w_{i,j}\, e_i f_j^\top, \quad \text{where } f_j^\top \text{ is row } j \text{ of } X.$$

Hence the compression metric for element-wise compressors in Transformer layers is

$$\boxed{\mathcal{L}_{\text{comp}}(w_{i,j}) \;=\; \| w_{i,j}\, e_i f_j^\top \|_F^2 \;=\; w_{i,j}^2\, \| f_j \|_F^2.}$$

---

**Algorithm 2:** Discrepancy-aware compression

---

**Input** : Layer update $W$; Compression granularity $g \in \{\text{ELEMENT, SINGULAR TRIPLET}\}$;
  Communication budget B (compression ratio or retained rank);
**Output:** compressed layer update $\widehat{W}$ under communication budget B.

---

1 **Define** compression unit set $\mathcal{U}$ (whether elements or singular triplets).
2 **for** *each* $u \in \mathcal{U}$ **do**
3 | Randomly select a small sample size of calibration data to obtain calibration activation $X$,
  | and compute compression metric $\mathcal{L}_{\text{comp}}(u) = \sum_{\text{cal } X} \|Y - Y'\|_F^2$.
4 **Select** a subset $\mathcal{S} \subseteq \mathcal{U}$ that satisfies the budget B and maximizes $\sum_{u \in \mathcal{S}} \mathcal{L}_{\text{comp}}(u)$
5 **Form** $\widehat{W}$ by retaining units in $\mathcal{S}$ and dropping the rest.

---

**For Element-wise Sparsification in Convolution Layers.** Consider an individual kernel with input channels $C_{in}$, output channels $C_{out}$, stride $s$, and padding $p$. Dropping a single kernel element $w_{k,c}[i,j]$ with $0 \leq i,j < F$ yields an output perturbation

$$\Delta Y_k[u,v] = w_{k,c}[i,j] \, \bar{X}_c[us+i, \, vs+j].$$

Hence the compression metric for element-wise compressors in Convolution layers is

$$\boxed{\mathcal{L}_{\text{comp}}(w_{k,c}[i,j]) = \|\Delta Y_k\|_F^2 = w_{k,c}[i,j]^2 \sum_{u=0}^{H'-1} \sum_{v=0}^{W'-1} \left(\bar{X}_c[us+i, \, vs+j]\right)^2.}$$

Here $H' = \lfloor \frac{H+2p-F}{s} \rfloor + 1$ and $W' = \lfloor \frac{W+2p-F}{s} \rfloor + 1$.

**For Low-rank Decomposition in Transformer Layers.** Let the SVD be $W = U\Sigma V^\top = \sum_{t=1}^r \sigma_t u_t v_t^\top$. Dropping a single singular value $\sigma_t$ causes

$$Y - Y' = (W - W_{\backslash t})X = \sigma_t \, u_t v_t^\top X,$$

Hence the compression metric for low-rank compressors in Transformer layers is

$$\boxed{\mathcal{L}_{\text{comp}}(\sigma_t) = \|\sigma_t \, u_t(v_t^\top X)\|_F^2 = \sigma_t^2 \|v_t^\top X\|_F^2.}$$

**For Low-rank Decomposition in Convolution Layers.** Fix $(k,c)$ and take the SVD of its kernel $W_{k,c} = \sum_{t=1}^r \sigma_t \, u_t v_t^\top$ with $r \leq F$ and unit singular vectors $u_t, v_t \in \mathbb{R}^F$. Dropping component $t$ removes its rank-1 contribution, and the resulting output perturbation with stride $s$ and padding size $p$ is given explicitly by

$$\Delta Y_k^{(t,c)}[m,n] = \sigma_t \sum_{i=0}^{F-1} \sum_{j=0}^{F-1} u_t[i] \, v_t[j] \, \bar{X}_c(ms+i, \, ns+j).$$

To evaluate efficiently, we stack $\{v_t^\top\}_{t=1}^r$ as $r$ horizontal $1 \times F$ filters and compute $Z_t = H_{v_t}^{(s,P)}(\bar{X}_c)$ in a single pass; then apply the $r$ vertical $F \times 1$ filters $\{u_t\}$ in grouped fashion to obtain $Y_t = V_{u_t}^{(s,P)}(Z_t)$ for all singular values. A naive implementation would perform $r$ separate $F \times F$ 2D correlations with cost $O(r \, F^2 \, H'W')$, whereas this two-pass scheme costs $O(r \, F \, H'W')$. Define

$$\left(H_v^{(s,p)}(\bar{X}_c)\right)[m',n] = \sum_{j=0}^{F-1} v[j] \, \bar{X}_c(m', \, ns+j), \; \left(V_u^{(s,p)}(Z)\right)[m,n] = \sum_{i=0}^{F-1} u[i] \, Z(ms+i, \, n),$$

so that $\Delta Y_k^{(t,c)}[m,n] = \sigma_t \left(V_{u_t}^{(s,p)}\left(H_{v_t}^{(s,p)}(\bar{X}_c)\right)\right)[m,n]$. Then the compression metric for low-rank compressors in Convolution layers is

$$\boxed{\mathcal{L}_{\text{comp}}(\sigma_t; k,c) = \|\Delta Y_k^{(t,c)}\|_F^2 = \sigma_t^2 \|V_{u_t}^{(s,p)}\left(H_{v_t}^{(s,p)}(\bar{X}_c)\right)\|_F^2.}$$

## 4 EXPERIMENTS

### 4.1 SETUP

We evaluate on CIFAR-10, CIFAR-100 (Krizhevsky et al., 2009), and Fashion-MNIST (Xiao et al., 2017) datasets under non-IID client partitions, using three Transformers (ViT-tiny, ViT-small, and ViT-base (Dosovitskiy et al., 2021)) and two CNNs (AlexNet (Krizhevsky et al., 2012) and ResNet-18 (He et al., 2016)). We simulate the non-IID scenario by considering a heterogeneous partition, where the number of data points and class proportions are unbalanced. Specifically, we simulate a partition into $N$ clients following a Dirichlet allocation with $\alpha = 0.2$. The level of heterogeneity among local datasets across different clients decreases as $\alpha$ increases. We set the total number of clients to be $N = 100$, and randomly sample 10 clients per round for training and aggregation. The training process consists of 200 communication rounds. Each participating client performs $E = 2$ local training epochs per communication round with batch size $bs = 16$. Training employs a cosine annealing learning rate schedule (Loshchilov & Hutter, 2017) and linear warm-up. For ViTs, we use the AdamW optimizer (Loshchilov & Hutter, 2019) with an initial learning rate of $10^{-4}$ decayed by a cosine annealing factor and a weight decay of 0.05, while for ResNet-18 we use the SGD optimizer with an initial learning rate of 0.01 decayed by a cosine annealing factor and a weight decay of $10^{-4}$. Unless stated otherwise, each client uses 64 randomly sampled local examples per round for calibration. All clients initialize from the same global model and employ identical within-round learning-rate schedules. We evaluate two baselines: the element-wise sparsification method Top-k and the low-rank decomposition method ATOMO. For each, we compare the original magnitude-based method with our discrepancy-aware re-ranking variant under the same communication budget. For element-wise sparsification, we use the global budget configuration, as it typically performs better and is more widely adopted than the layer budget. For low-rank decomposition, we follow ATOMO's original setup with the layer budget configuration (Wang et al., 2018). We conduct each experiment with three independent trials and report the average results. In all experiments, we employ the classic error feedback scheme to mitigate the bias introduced by lossy compression.

### 4.2 MAIN RESULTS

**Element-wise Sparsification.** To evaluate the effectiveness of our discrepancy-aware compression in enhancing existing element-level sparsification methods, we compare the final test accuracy of the original magnitude-based Top-k method and its discrepancy-aware augmented variant across different compression ratios, datasets, and models. As shown in Table 1, the discrepancy-aware compression leads to consistent improvements over the baseline. Notably, the advantage becomes more pronounced as the compression ratio decreases. This is because under stronger compression, the overlap between compression units selected by different methods reduces, as detailed in Section 4.3. Thus, the ability to identify important information becomes more critical to performance.

Table 1: Final test accuracy of the element-wise sparsification method Top-k and the corresponding discrepancy-aware augmented method across different compression ratios, datasets, and models.

| Dataset (Model) | Method | Compression Ratio | | | | | |
|---|---|---|---|---|---|---|---|
| | | 0.01 | 0.1 | 0.2 | 0.4 | 0.6 | 1.0 |
| CIFAR-10 (ViT-tiny) | Magnitude-based | $21.03_{\pm 0.3}$ | $34.93_{\pm 0.5}$ | $37.69_{\pm 0.3}$ | $38.25_{\pm 0.2}$ | $38.57_{\pm 0.2}$ | $51.56_{\pm 0.3}$ |
| | Discrepancy-aware | $\mathbf{29.62}_{\pm 0.1}$ | $\mathbf{41.52}_{\pm 0.4}$ | $\mathbf{43.11}_{\pm 0.4}$ | $\mathbf{42.81}_{\pm 0.1}$ | $\mathbf{41.21}_{\pm 0.2}$ | |
| CIFAR-100 (ViT-small) | Magnitude-based | $9.29_{\pm 0.4}$ | $26.91_{\pm 0.3}$ | $28.51_{\pm 0.5}$ | $29.97_{\pm 0.4}$ | $30.23_{\pm 0.5}$ | $34.13_{\pm 0.2}$ |
| | Discrepancy-aware | $\mathbf{13.29}_{\pm 0.2}$ | $\mathbf{28.62}_{\pm 0.3}$ | $\mathbf{31.27}_{\pm 0.1}$ | $\mathbf{32.87}_{\pm 0.4}$ | $\mathbf{33.32}_{\pm 0.3}$ | |
| CIFAR-100 (ResNet-18) | Magnitude-based | $10.76_{\pm 0.1}$ | $27.28_{\pm 0.7}$ | $30.71_{\pm 1.0}$ | $32.34_{\pm 0.9}$ | $32.82_{\pm 0.6}$ | $35.33_{\pm 0.1}$ |
| | Discrepancy-aware | $\mathbf{15.58}_{\pm 0.3}$ | $\mathbf{29.71}_{\pm 0.4}$ | $\mathbf{32.54}_{\pm 0.5}$ | $\mathbf{33.65}_{\pm 0.7}$ | $\mathbf{33.71}_{\pm 0.4}$ | |
| Fashion-MNIST (AlexNet) | Magnitude-based | $63.31_{\pm 0.1}$ | $70.32_{\pm 0.9}$ | $71.50_{\pm 0.5}$ | $71.59_{\pm 0.7}$ | $71.98_{\pm 0.7}$ | $78.96_{\pm 0.3}$ |
| | Discrepancy-aware | $\mathbf{67.55}_{\pm 0.3}$ | $\mathbf{73.42}_{\pm 0.7}$ | $\mathbf{73.61}_{\pm 0.4}$ | $\mathbf{73.70}_{\pm 0.4}$ | $\mathbf{74.01}_{\pm 0.5}$ | |

**Low-rank Decomposition.** Similarly, we evaluate discrepancy-aware compression applied to low-rank decomposition methods by comparing the original ATOMO method and its discrepancy-aware augmented variant across various retained ranks, datasets, and models. As shown in Table 2,

Table 2: Final test accuracy of the low-rank decomposition method ATOMO and the corresponding discrepancy-aware augmented method across different retained ranks, datasets, and models.

| Dataset (Model) | Method | Retained Rank | | | |
|---|---|---|---|---|---|
| | | 1 | 2 | 4 | 8 |
| CIFAR-10 (ViT-tiny) | Magnitude-based | $26.46_{\pm 0.4}$ | $28.33_{\pm 0.3}$ | $36.14_{\pm 0.4}$ | $42.44_{\pm 0.3}$ |
| | Discrepancy-aware | $\mathbf{28.03}_{\pm 0.5}$ | $\mathbf{30.74}_{\pm 0.3}$ | $\mathbf{36.40}_{\pm 0.4}$ | $\mathbf{42.68}_{\pm 0.2}$ |
| CIFAR-10 (ViT-small) | Magnitude-based | $33.41_{\pm 0.4}$ | $37.01_{\pm 0.4}$ | $41.32_{\pm 0.2}$ | $44.01_{\pm 0.4}$ |
| | Discrepancy-aware | $\mathbf{34.61}_{\pm 0.4}$ | $\mathbf{40.10}_{\pm 0.3}$ | $\mathbf{43.17}_{\pm 0.3}$ | $\mathbf{45.29}_{\pm 0.3}$ |
| CIFAR-100 (ViT-base) | Magnitude-based | $17.08_{\pm 0.4}$ | $19.07_{\pm 0.5}$ | $24.62_{\pm 0.7}$ | $29.01_{\pm 0.6}$ |
| | Discrepancy-aware | $\mathbf{20.17}_{\pm 0.3}$ | $\mathbf{23.99}_{\pm 0.5}$ | $\mathbf{26.13}_{\pm 0.4}$ | $\mathbf{30.62}_{\pm 0.4}$ |

the discrepancy-aware augmented approach again leads to consistent and significant improvements. As with element-wise sparsification, the benefit increases under higher compression, due to reduced overlap and increased differentiation among selected candidate compression units—making accurate identification of important features more critical, as further analyzed in Section 4.3.

**Overall Communication Rounds to Achieve Target Accuracy.** To demonstrate that the discrepancy-aware method converges in fewer communication rounds, we further compare the number of communication rounds required to reach target accuracy between magnitude-based and discrepancy-aware compression methods. Specifically, we conduct experiments on two representative configurations at various compression ratios. The target accuracy is defined as 60% or 80% of the final test accuracy achieved by the discrepancy-aware method for each specific configuration. The results are summarized in Tables 3 and 4. Across all configurations and compression ratios, the discrepancy-aware method consistently reaches the same target accuracy in fewer rounds than the magnitude-based baseline. The performance gains are particularly pronounced under strong compression, with a speedup of up to 1.56× in reaching the target accuracy at a compression ratio of 0.01 on the CIFAR-10 dataset using the ViT-tiny model. This analysis demonstrates that discrepancy-aware compression not only enhances final accuracy, but also accelerates convergence in terms of wall-clock time under the same communication budget.

Table 3: Communication rounds required to reach target accuracy on FMNIST (AlexNet) of the element-wise sparsification method Top-k and the corresponding discrepancy-aware variant. Values in parentheses indicate the speedup of discrepancy-aware methods over magnitude-based baselines.

| Target Accuracy | Method | Compression Ratio | | | | | |
|---|---|---|---|---|---|---|---|
| | | 0.01 | 0.05 | 0.1 | 0.2 | 0.4 | 0.6 |
| 60% of the final test accuracy | **Magnitude-based** | 73 | 54 | 55 | 54 | 56 | 56 |
| | **Discrepancy-aware** | 62(**1.18**×) | 46(**1.17**×) | 50(**1.10**×) | 54(1.00×) | 56(1.00×) | 53(**1.05**×) |
| 80% of the final test accuracy | **Magnitude-based** | 105 | 85 | 83 | 83 | 84 | 81 |
| | **Discrepancy-aware** | 82(**1.28**×) | 72(**1.18**×) | 71(**1.17**×) | 83(1.00×) | 81(**1.03**×) | 82(0.98×) |

Table 4: Communication rounds required to reach target accuracy on CIFAR-10 (ViT-tiny) of the element-wise sparsification method Top-k and the corresponding discrepancy-aware variant. Values in parentheses indicate the speedup of discrepancy-aware methods over magnitude-based baselines.

| Target Accuracy | Method | Compression Ratio | | | | | |
|---|---|---|---|---|---|---|---|
| | | 0.01 | 0.05 | 0.1 | 0.2 | 0.4 | 0.6 |
| 60% of the final test accuracy | **Magnitude-based** | 53 | 45 | 33 | 36 | 39 | 40 |
| | **Discrepancy-aware** | 44(**1.20**×) | 31(**1.45**×) | 26(**1.27**×) | 34(**1.06**×) | 37(**1.05**×) | 39(**1.03**×) |
| 80% of the final test accuracy | **Magnitude-based** | 128 | 117 | 120 | 99 | 116 | 89 |
| | **Discrepancy-aware** | 82(**1.56**×) | 81(**1.44**×) | 83(**1.45**×) | 79(**1.25**×) | 79(**1.47**×) | 86(**1.03**×) |

## 4.3 OVERLAP ANALYSIS OF SELECTED TRANSMITTED CONTENT

To demonstrate the differences between discrepancy-aware and magnitude-based selections across varying compression levels and data heterogeneity, we report the overlap rate of the selected compression units between the Top-k method and its discrepancy-aware augmented variant. This is shown in the 2D heatmaps of Figure 1, with the compression ratio (x-axis) and Dirichlet parameter

$\alpha$ (y-axis). Across datasets and models, the overlap increases with both the compression ratio and $\alpha$. When compression is aggressive and data are highly non-IID, the overlap is minimal, whereas milder compression and more IID data yield substantially higher overlap. These trends reinforce our central conclusion: discrepancy-aware compression is most beneficial under conditions of tight bandwidth with low compression ratios and strong heterogeneity with small $\alpha$. When resources are constrained, discrepancy-aware compression selectively transmits the most important content. Additionally, the uneven distribution of local data characteristics across clients increases the impact of these characteristics on the importance of compression units. However, as compression becomes milder and data approach IID, the two criteria partially align and the overlap increases. This suggests that while large-magnitude units are no longer the most critical, they still regain a relative advantage. We also observe higher overlaps on Fashion-MNIST (AlexNet) compared to CIFAR-10 (ViT-tiny), indicating that simpler datasets and architectures lead to more agreement between the two strategies.

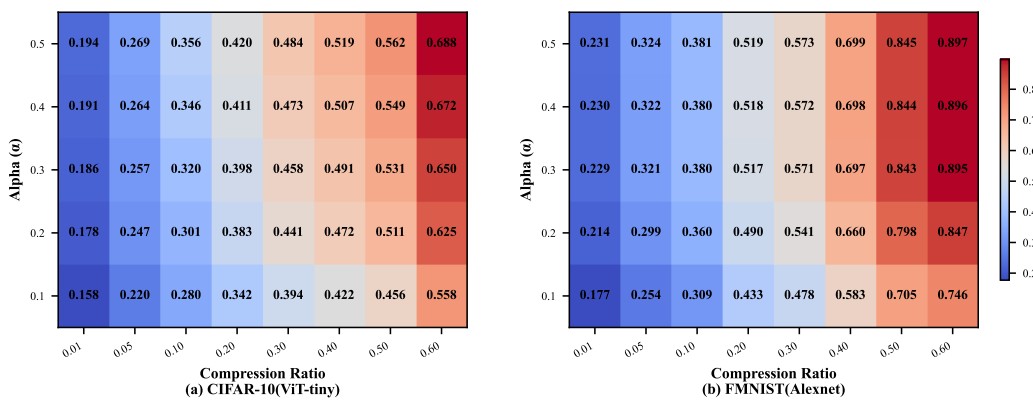

Figure 1: Overlap rate heatmaps for different Non-IID degree (alpha) and compression ratios on (a) CIFAR-10 (ViT-tiny) and (b) FMNIST (AlexNet).

## 4.4 ANALYSIS AND ABLATION

**Selection Strategies and Sample Size of Calibration Data.** We investigate the influence of calibration data selection strategies and the sample size of calibration data on the performance of discrepancy-aware compression. As shown in Table 5, randomly sampling calibration data at each round consistently outperforms strategies that always choose either the earliest or the latest samples, without incurring additional cost. Therefore, we adopt random sampling as the default strategy in experiments. Moreover, Table 5 demonstrates that the number of calibration samples has only a marginal effect on performance, which remains stable across different quantities. These findings highlight the robustness of our method in different calibration sample sizes.

Table 5: Impact of calibration data selection strategies and sample size.

| Dataset (Model) | Selection Strategies | | | Calibration Samples | | | | |
|---|---|---|---|---|---|---|---|---|
| | Random | First | Last | 32 | 64 | 128 | 256 | 512 |
| Fashion-MNIST (AlexNet) | 73.95 | 72.49 | 73.14 | 72.89 | 72.41 | 72.86 | 72.76 | 72.91 |
| CIFAR-10 (ViT-tiny) | 43.23 | 43.14 | 42.91 | 42.97 | 43.11 | 43.52 | 43.22 | 42.96 |

**Additional computational cost.** We evaluated the additional computational cost introduced by our discrepancy-aware compression method by measuring the average time spent at different stages. As shown in Table 6, while the compression time with the discrepancy-aware method increased by up to $18.2\%$, the overall computation time per client per round showed only a slight increase. Given the low communication frequency and communication bottlenecks in FL, it is an acceptable trade-off. Our method significantly enhances communication resource utilization under tight communication budgets and thereby improves performance. This makes it a viable strategy for enhancing communication efficiency in FL under constrained communication resources.

Table 6: Comparison of per-round time consumption per client (calibration sample size = 64).

| Dataset (Model) | Method | Avg Time Consumption (Top-k) | | | Avg Time Consumption (ATOMO) | | |
|---|---|---|---|---|---|---|---|
| | | Training | Compression | Total | Training | Compression | Total |
| CIFAR-10 (ViT-tiny) | Magnitude-based | 0.5681s | 0.0013s | 0.5694s | 0.7033s | 0.1006s | 0.8039s |
| | Discrepancy-aware | 0.5662s | 0.0568s | 0.6380s | 0.7055s | 0.2232s | 0.9287s |
| CIFAR-100 (ViT-small) | Magnitude-based | 0.5789s | 0.0014s | 0.5803s | 0.7480s | 0.3258s | 1.0738s |
| | Discrepancy-aware | 0.5799s | 0.0586s | 0.6385s | 0.7423s | 0.5217s | 1.2694s |

**Non-IID Degree.** We simulate a heterogeneous data partition into $N$ clients using the Dirichlet distribution with $\alpha$. Here, we conduct an ablation study to investigate the impacts of Non-IID degree, while keeping all other settings identical to Section 4.1. As shown in Figure 2, across all heterogeneity settings, discrepancy-aware compression consistently improves the final test accuracy over magnitude-based baselines. Notably, the gains are most pronounced under highly non-IID splits with smaller $\alpha$, suggesting that selecting candidate compression units according to client-specific calibration data, rather than pure magnitude, can better capture what matters for each client. Some large-magnitude candidate compression units exert little influence on the local input–output mapping and are therefore deprioritized by discrepancy-aware compression. As the non-IID degree decreases as $\alpha$ increases, the final test accuracy generally improves as expected.

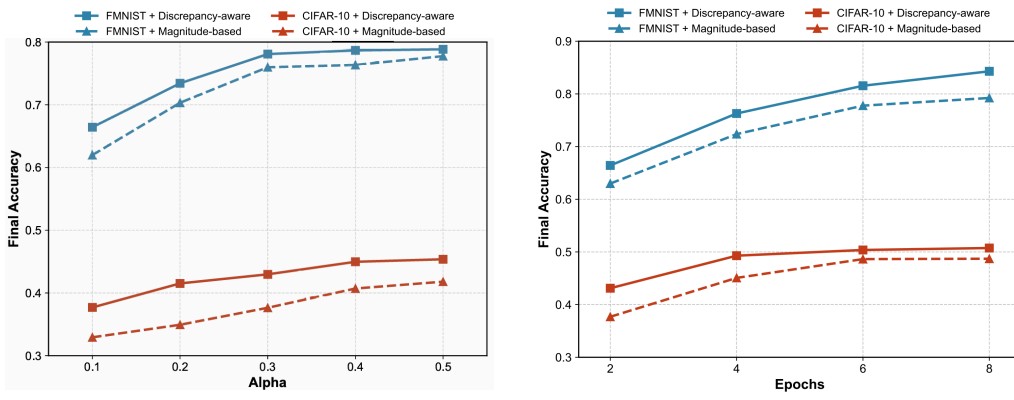

Figure 2: Impact of non-IID degree.   Figure 3: Impact of local training epochs.

**Local Training Epochs.** We conduct an ablation study to investigate the impact of local training epochs, while keeping all other settings identical to Section 4.1. As shown in Figure 3, increasing the number of local training epochs consistently improves the final test accuracy, reflecting the benefit of more thorough local updates. Across all settings, our discrepancy-aware compression achieves higher accuracy than magnitude-based baselines. These results highlight that selecting compression units based on calibration data consistently provides robust gains.

## 5 CONCLUSION

In this work, we introduce a discrepancy-aware communication compression strategy for FL that addresses the challenge under tight communication budgets by minimizing the output discrepancy through calibration data. This approach can be seamlessly integrated with existing FL compression techniques, offering enhanced performance under limited communication resources. Moreover, it is compatible with a wide range of other FL methods, including those focused on security, privacy, and heterogeneity. Empirical results across various datasets and models demonstrate that the proposed method significantly outperforms baseline compression strategies. Our work provides a fresh perspective on communication compression process in FL, and we believe this work can significantly contribute to the development of communication-efficient FL.

## 6 ACKNOWLEDGMENT

This work was supported by the National Key Research and Development Program of China under Grant 2023YFB4301900 and Beijing Nova Program 2024102

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

## A APPENDIX

### A.1 LLM USAGE DECLARATION

Large language models (e.g., ChatGPT) were used solely for language editing and formatting. They did not contribute to the conception, design, implementation, analysis, data generation or labeling, or evaluation of the methods and results. All technical content and claims were authored and verified by the authors, and no personal, proprietary, or sensitive data were shared with LLM services.

## A.2 COMPARISON OF CONTENT TO BE COMPRESSED IN FEDERATED LEARNING AND DISTRIBUTED LEARNING

Let the global objective be

$$F(W) = \sum_{k=1}^{K} p_k \, \mathbb{E}_{\xi \sim \mathcal{D}_k} \big[ \ell(W; \xi) \big],$$

and $g(W) = \nabla F(W)$ represents the gradient, and $H(W) = \nabla^2 F(W)$ is the Hessian.

In data-center distributed learning (DL), every iteration synchronizes across workers, transmitting gradients, and the compression step involves reducing the gradient $g(W)$ using a compressor $C(\cdot)$, such as Top-k. For a given parameter update $W$, the compressed update $\widehat{\Delta W} = C(\Delta W)$ leads to a compression loss:

$$F(W + \widehat{\Delta W}) - F(W + \Delta W) = \eta g^\top (g - C(g)) + O(\eta^2),$$

where $C(g)$ zeroes out some components of the gradient. In this case, the compression loss is directly related to the gradient magnitude, specifically to $g_j^2$, because the object being compressed and transmitted is the gradient itself.

However, in FL, the situation is different. Instead of transmitting a single gradient, each client performs multiple local SGD steps, accumulating updates over several epochs. The update for client $k$ is the difference:

$$\Delta W_k = W_{k,t} - W_{k,0} = -\eta \sum_{\tau=0}^{t-1} g_k(W_{k,\tau}),$$

which is a series of gradients evaluated at different local iterates. Expanding around the global model $W$, we approximate:

$$g_k(W_{k,\tau}) \approx g_k(W) + H_k(W)(W_{k,\tau} - W).$$

Since $W_{k,\tau} - W$ is a summation of previous gradients, the update becomes:

$$\Delta W_k \approx -\eta t g_k(W) + \eta^2 \frac{t(t-1)}{2} H_k(W) g_k(W),$$

where the first term is the gradient-based update and the second term introduces curvature effects due to the Hessian. This shows that the client updates are influenced by both the gradient and the model curvature $H_k(W)$, making the relationship between the update and the gradient more complex than in DL.

Now consider the server-side aggregation of updates from multiple clients:

$$\Delta W = \sum_k \alpha_k \Delta W_k.$$

The compressed update is $\widehat{\Delta W} = C(\Delta W)$, and the compression error is given by the residual $R = \Delta W - \widehat{\Delta W}$. The resulting compression loss is:

$$F(W + \widehat{\Delta W}) - F(W + \Delta W) = -g(W)^\top R + O(\|\Delta W\| \|R\| + \|R\|^2).$$

Crucially, this residual $R$ is a difference in parameter updates, not gradients. The importance of a coordinate $j$ is determined by the product $g_j(W) R_j$, not just by the magnitude $|R_j|$ alone. This reflects a fundamental difference from DL, where magnitude-based compression decisions align with gradient importance.

This analysis confirms the core observation: in FL, the content being compressed (parameter updates) carries little gradient information, which weakens the relationship between magnitude and importance. This motivates our discrepancy-aware compression strategy, which ranks compression units based on their effect on layer output using the $\mathcal{L}_{\text{comp}}$ rather than relying on magnitude or randomness.

A.3 COMPATIBILITY WITH POWERSGD

We have already shown in the main text that our discrepancy-aware selection principle can be instantiated on top of ATOMO, where the layer update is factorized via an exact SVD. Importantly, the same principle is not restricted to this setting. Our discrepancy-aware selection mechanism only requires that a layer update matrix can be written as a sum of rank-1 components and does not rely on these components being exact singular vectors. Concretely, suppose a layer update matrix $W \in \mathbb{R}^{d_{\text{out}} \times d_{\text{in}}}$ admits a factorization of the form

$$W \approx \sum_{t=1}^{r} a_t b_t^\top, \tag{1}$$

where $a_t \in \mathbb{R}^{d_{\text{out}}}$ and $b_t \in \mathbb{R}^{d_{\text{in}}}$ define rank-1 components $a_t b_t^\top$. Let $X \in \mathbb{R}^{d_{\text{in}} \times m}$ denote a small local calibration set of layer inputs (activations). The corresponding layer outputs are

$$Y = (W_0 + W)X, \tag{2}$$

where $W_0$ is the current model parameter of the layer. If we remove a single component $a_t b_t^\top$, the modified update becomes $W - a_t b_t^\top$, and the new outputs are

$$Y' = (W_0 + W - a_t b_t^\top)X = Y - a_t b_t^\top X. \tag{3}$$

Hence, the discrepancy incurred on the calibration set by dropping component $t$ is

$$\Delta_t = Y - Y' = a_t b_t^\top X = a_t (b_t^\top X). \tag{4}$$

Our per-component discrepancy metric is precisely the Frobenius norm of this difference:

$$L_{\text{comp}}(t) = \|\Delta_t\|_F^2 = \left\| a_t b_t^\top X \right\|_F^2 = \left\| a_t (b_t^\top X) \right\|_F^2. \tag{5}$$

In the ATOMO instantiation, $W$ is factorized using the SVD as $W = \sum_{t=1}^{r} \sigma_t u_t v_t^\top$. In this special case, $(a_t, b_t) = (\sigma_t u_t, v_t)$, and $L_{\text{comp}}(t)$ reduces to a function of the singular value and the right singular vector evaluated on the calibration inputs.

**Specialization to PowerSGD.** PowerSGD produces a low-rank approximation of the form

$$W \approx PQ^\top, \tag{6}$$

where $P \in \mathbb{R}^{d_{\text{out}} \times r}$ and $Q \in \mathbb{R}^{d_{\text{in}} \times r}$ are obtained via iterative power iterations. Let $p_t$ and $q_t$ denote the $t$-th columns of $P$ and $Q$, respectively. Then

$$PQ^\top = \sum_{t=1}^{r} p_t q_t^\top, \tag{7}$$

so $(a_t, b_t) = (p_t, q_t)$ in Equation 5. The discrepancy metric for component $t$ under PowerSGD becomes

$$L_{\text{comp}}(t) = \left\| p_t q_t^\top X \right\|_F^2 = \left\| p_t (q_t^\top X) \right\|_F^2, \quad t = 1, \dots, r. \tag{8}$$

Thus, our method applies directly to the rank-1 components produced by PowerSGD; it does not depend on how $P$ and $Q$ are computed, only on the resulting factorization $W \approx PQ^\top$.

**Discrepancy-aware selection on top of PowerSGD.** Given a standard PowerSGD implementation for a layer, our discrepancy-aware selection can be implemented in each communication round as follows.

1. **PowerSGD factorization with an oversized candidate rank.** Run PowerSGD to obtain a rank-$r_{\text{cand}}$ approximation $W \approx PQ^\top$ for each compressed layer, where $r_{\text{cand}}$ is a pre-specified *candidate rank* that is intentionally larger than the effective rank allowed by the communication budget, i.e., $r_{\text{budget}} < r_{\text{cand}}$.

2. **Compute discrepancy scores on a calibration set.** On a small local calibration set $X$, compute $q_t^\top X$ for each column $q_t$ of $Q$, and evaluate

$$L_{\text{comp}}(t) = \left\| p_t (q_t^\top X) \right\|_F^2, \quad t = 1, \dots, r_{\text{cand}}. \tag{9}$$

3. **Select components under the communication budget.** Rank the $r_{\text{cand}}$ components $\{p_t q_t^\top\}$ by $L_{\text{comp}}(t)$ in descending order, and select the top $r_{\text{budget}}$ components that fit within the desired communication budget.

4. **Form truncated factors and communicate them.** Form truncated factors $P' \in \mathbb{R}^{d_{\text{out}} \times r_{\text{budget}}}$ and $Q' \in \mathbb{R}^{d_{\text{in}} \times r_{\text{budget}}}$ by keeping only the selected columns of $P$ and $Q$, and communicate $(P', Q')$ instead of the full $(P, Q)$.

This procedure does not modify the internal power-iteration scheme or the encoding format of PowerSGD; it only changes *which* rank-1 components are ultimately transmitted, based on their data-driven impact on the layer output as measured by $L_{\text{comp}}(t)$.

## A.4 FUTURE WORK: EXTENSION TO QUANTIZATION-BASED METHODS

Our discrepancy-aware selection principle is designed to be compressor-agnostic: it only requires the ability to simulate the effect of a given compression operation on the layer outputs over a small calibration set. In the main text, this principle is instantiated for element-wise sparsification (e.g., Top-$k$) and low-rank decomposition (e.g., ATOMO) by treating either an individual element or a singular triplet as the compression unit and scoring it via the induced output discrepancy.

The same idea naturally extends to quantization-based methods. Consider a set of quantization units $\mathcal{U}$. For each unit $u \in \mathcal{U}$, we allow a finite set of candidate bit-widths $\mathcal{B}$. For a given pair $(u, b)$ with $b \in \mathcal{B}$, we construct a compression configuration $\theta(u, b)$ where unit $u$ is quantized with $b$ bits, and all other units $u' \neq u$ are kept at a fixed reference precision (e.g., full precision or a baseline bit-width). We then measure the corresponding discrepancy

$$L(u,b) \;=\; L(\theta(u,b)) \;=\; \big\|(W_0 + W)X - (W_0 + \mathcal{C}_{\theta(u,b)}(W))X\big\|_F^2. \tag{10}$$

Given a fixed total bit budget for each layer, bits can be allocated by prioritizing units for which increasing the bit-width yields the largest reduction in discrepancy per additional bit. A simple strategy is to start from a baseline bit-width $b_{\text{base}}$ for all units and then iteratively consider raising the bit-width of some units to higher candidate values. For each unit $u$ and a feasible higher bit-width $b' > b_{\text{curr}}(u)$, we estimate the decrease in the discrepancy measure $L(u,b)$ when changing from $b_{\text{curr}}(u)$ to $b'$, and normalize this decrease by the corresponding increase in the number of bits. At each step, we choose the unit and bit-width change that provides the largest discrepancy reduction per additional bit, update $b_{\text{curr}}(u)$ accordingly, and repeat this greedy procedure until the total bit budget for the layer is fully used. This yields a discrepancy-aware quantization scheme in which units that are more critical for the layer outputs (high-discrepancy units) are assigned more bits, while less critical ones can be more aggressively quantized.

This construction is complementary to existing federated learning quantization frameworks: the discrepancy-aware principle can be used as a drop-in mechanism for deciding bit allocation within such frameworks, without modifying their overall communication or system design. We expect that specific instantiations of this idea, tailored to particular quantization schemes and hardware constraints, will be promising directions for future work in communication-efficient quantization.

## A.5 CONVERGENCE OF DISCREPANCY-AWARE COMPRESSOR WITH ERROR-FEEDBACK

In this section, following previous works on compressor convergence in distributed learning (Beznosikov et al., 2023; Karimireddy et al., 2019; Condat et al., 2022; Hu et al., 2021b; Qian et al., 2021; Li & Li, 2023), we adopted similar settings and approaches to discuss the convergence of our discrepancy-aware compressor.

We consider minimizing a possibly non-convex objective

$$\min_{x \in \mathbb{R}^d} f(x) := \mathbb{E}_\xi\big[F(x; \xi)\big]. \tag{11}$$

### A.5.1 ERROR–FEEDBACK SCHEME

We study the following generic error–feedback scheme, as detailed in Algorithm 3.

---

**Algorithm 3:** Error–feedback SGD with Compressor

---

**Input:** stepsize $\gamma > 0$, compressor $C(\cdot)$, initial weight matrix $w^0$;

1 Set $e_0 = 0$. **for** $t = 0, 1, \ldots, T$ **do**
2      Draw stochastic gradient $g_t$ at $w_t$.
3      Form corrected direction $g'_t = \gamma g_t + e_t$.
4      Compress: $C(g'_t)$.
5      Update iterate: $w_{t+1} = w_t - C(g'_t)$.
6      Update error: $e_{t+1} = g'_t - C(g'_t)$.

---

### A.5.2 ASSUMPTIONS

**Assumption 1** (Smoothness). *The function $f$ is $L$–smooth, i.e. for all $x, y \in \mathbb{R}^d$,*

$$f(y) \leq f(x) + \langle \nabla f(x), y - x \rangle + \frac{L}{2} \|y - x\|^2. \tag{12}$$

**Assumption 2** (Unbiased stochastic gradients with bounded second moment). *At iteration $t$ the algorithm queries a stochastic gradient $g_t$ such that*

$$\mathbb{E}[g_t \mid w_t] = \nabla f(w_t), \qquad \mathbb{E}\big[\|g_t\|^2 \mid w_t\big] \leq \sigma^2, \tag{13}$$

*for some $\sigma > 0$.*

**Assumption 3** ($\delta$–approximate compressor). *A (possibly randomized) operator $C : \mathbb{R}^d \to \mathbb{R}^d$ is called a $\delta$–approximate compressor with $\delta \in (0, 1]$ if for all $v \in \mathbb{R}^d$*

$$\mathbb{E}\big[\|C(v) - v\|^2\big] \leq (1 - \delta) \|v\|^2. \tag{14}$$

*The expectation is taken over the internal randomness of $C$ if any.*

### A.5.3 CONVERGENCE ANALYSIS

**Lemma 1** (discrepancy-aware top-$k$ compressor is a $\delta$-approximate compressor). *Let $a \in \mathbb{R}^d$ denote the fixed calibration activations obtained from the calibration dataset, and let $m_i(w, a_i)$ be a scalar discrepancy score for coordinate $i$ which depends on both the weight entry $w_i$ and the calibration activation $a_i$. We assume that $m_i(\cdot, a_i)$ is continuous and strictly increasing in $|w_i|$ for every fixed $a_i$. The discrepancy-aware top-k compressor $C_{\mathrm{dis}-\mathrm{k}}$ is defined by*

$$S_{\mathrm{dis}-\mathrm{k}\text{-}k}(w) := \text{indices of the } k \text{ largest } m_i(w, a_i), \qquad (C_{\mathrm{dis}-\mathrm{k}}(w))_i := \begin{cases} w_i, & i \in S_{\mathrm{dis}-\mathrm{k}\text{-}k}(w), \\ 0, & \text{otherwise.} \end{cases} \tag{15}$$

*Denote by $S_{\mathrm{top}\text{-}k}(w)$ the index set of the $k$ largest entries of $w$ in magnitude, and define*

$$\phi(w) := \frac{\sum_{i \in S_{\mathrm{dis}-\mathrm{k}\text{-}k}(w)} w_i^2}{\sum_{i \in S_{\mathrm{top}\text{-}k}(w)} w_i^2}, \qquad w \neq 0. \tag{16}$$

*Assume that on the feasible set $Q$ we have a uniform lower bound*

$$\phi(x) \geq c > 0, \qquad \forall x \in Q \setminus \{0\}, \tag{17}$$

*i.e., the subset selected by the discrepancy-aware rule always retains at least a fixed fraction $c$ of the squared $\ell_2$-mass that would be retained by plain top-$k$.*

*This assumption is mild for our discrepancy-aware compressor. During training, both the model parameters and the calibration activations are constrained to stay in a bounded region, so it is natural to restrict attention to a compact feasible set $Q \subset \mathbb{R}^d$ on which all coordinates are uniformly bounded. Moreover, for our $L_{\mathrm{comp}}$ the discrepancy score $m_i(w_i, a_i)$ depends not only on the calibration activation $a_i$. In particular, $|w_i|$ enters as an important multiplicative factor in the score. Hence the discrepancy-aware selection rule is hard to systematically assign large scores to coordinates with vanishingly small $|w_i|$ while ignoring coordinates with moderate or large $|w_i|$.*

*Then $C_{\mathrm{dis}-\mathrm{k}}$ is a $\delta$-approximate compressor on $Q$ with*

$$\delta = c\frac{k}{d}. \tag{18}$$

**Proof.** Fix any $w \in Q$. Since $C_{\text{dis}-\text{k}}(w)$ is obtained by zeroing out a subset of coordinates of $w$, we have

$$\|C_{\text{dis}-\text{k}}(w)\|_2^2 = \sum_{i \in S_{\text{dis}-\text{k}}(w)} w_i^2, \qquad \|w\|_2^2 = \sum_{i \in S_{\text{dis}-\text{k}}(w)} w_i^2 + \sum_{i \notin S_{\text{dis}-\text{k}}(w)} w_i^2, \qquad (19)$$

and therefore

$$\|C_{\text{dis}-\text{k}}(w) - w\|_2^2 = \sum_{i \notin S_{\text{dis}-\text{k}}(w)} w_i^2 = \|w\|_2^2 - \|C_{\text{dis}-\text{k}}(w)\|_2^2. \qquad (20)$$

On the other hand, the standard top-$k$ compressor satisfies (see Appendix B.2 in Karimireddy et al. (2019))

$$\sum_{i \in S_{\text{top-}k}(w)} w_i^2 \geq \frac{k}{d} \|w\|_2^2, \qquad (21)$$

i.e., the top-$k$ operator retains at least a $\frac{k}{d}$ fraction of the total energy. Together with Equation 17, this implies

$$\sum_{i \in S_{\text{dis}-\text{k}}(w)} w_i^2 = \phi(w) \sum_{i \in S_{\text{top-}k}(w)} w_i^2 \geq c \sum_{i \in S_{\text{top-}k}(w)} w_i^2 \geq c \frac{k}{d} \|w\|_2^2. \qquad (22)$$

Substituting Equation 22 into Equation 20, we obtain

$$\|C_{\text{dis}-\text{k}}(w) - w\|_2^2 = \|w\|_2^2 - \sum_{i \in S_{\text{dis}-\text{k}}(w)} w_i^2 \leq \left(1 - c\frac{k}{d}\right) \|w\|_2^2. \qquad (23)$$

Therefore, if we define

$$\delta := c\frac{k}{d}, \qquad (24)$$

then

$$\|C_{\text{dis}-\text{k}}(w) - w\|_2^2 \leq (1 - \delta) \|w\|_2^2, \qquad \forall w \in Q. \qquad (25)$$

Note that our compressor is deterministic, so the expectation in Assumption 3 over the internal randomness of $C$ is not needed in this case. Hence $C_{\text{dis}-\text{k}}$ is a $\delta$-approximate compressor on $Q$, which completes the proof.

**Lemma 2** (Bounded error). *Based on Assumptions 2 and Lemma 1, let $\{e_t\}_{t \geq 0}$ be generated by Algorithm 3. Then for any $t \geq 0$,*

$$\mathbb{E}\big[\|e_t\|^2\big] \leq \frac{4(1-\delta)}{\delta^2} \gamma^2 \sigma^2. \qquad (26)$$

**Proof.** By construction of $e_{t+1}$ and Assumption 3, we have

$$\|e_{t+1}\|^2 = \|g_t' - C(g_t')\|^2 \leq (1 - \delta) \|g_t'\|^2 = (1 - \delta) \|\gamma g_t + e_t\|^2. \qquad (27)$$

Taking expectation and expanding the square, for any parameter $\eta > 0$,

$$\mathbb{E}\big[\|e_{t+1}\|^2\big] \leq (1 - \delta) \mathbb{E}\big[\|\gamma g_t + e_t\|^2\big] \qquad (28)$$

$$\leq (1 - \delta)(1 + \eta) \mathbb{E}\big[\|e_t\|^2\big] + (1 - \delta)\Big(1 + \frac{1}{\eta}\Big)\gamma^2 \mathbb{E}\big[\|g_t\|^2\big] \qquad (29)$$

$$\leq (1 - \delta)(1 + \eta) \mathbb{E}\big[\|e_t\|^2\big] + (1 - \delta)\Big(1 + \frac{1}{\eta}\Big)\gamma^2 \sigma^2, \qquad (30)$$

where we used the inequality $\|a + b\|^2 \leq (1 + \eta)\|a\|^2 + (1 + 1/\eta)\|b\|^2$.

Unrolling this recursion and using $\mathbb{E}\|e_0\|^2 = 0$ gives

$$\mathbb{E}\big[\|e_{t+1}\|^2\big] \leq (1 - \delta)\Big(1 + \frac{1}{\eta}\Big)\gamma^2 \sigma^2 \sum_{i=0}^{t} \big[(1 - \delta)(1 + \eta)\big]^i. \qquad (31)$$

For any $\eta \in (0, \delta/(2(1-\delta)))$ we have $(1-\delta)(1+\eta) < 1$, and thus

$$\sum_{i=0}^{\infty} \left[(1-\delta)(1+\eta)\right]^i = \frac{1}{1-(1-\delta)(1+\eta)} = \frac{1}{\delta - \eta(1-\delta)}. \tag{32}$$

Choosing $\eta = \frac{\delta}{2(1-\delta)}$ yields

$$1 + \frac{1}{\eta} = 1 + \frac{2(1-\delta)}{\delta} \leq \frac{2}{\delta}, \qquad \delta - \eta(1-\delta) = \frac{\delta}{2},$$

and therefore

$$\mathbb{E}\big[\|e_{t+1}\|^2\big] \leq \frac{4(1-\delta)}{\delta^2} \gamma^2 \sigma^2. \tag{33}$$

Since the bound does not depend on $t$, it holds for all $t \geq 0$.

To relate Algorithm 3 to standard analysis in Karimireddy et al. (2019), it is convenient to introduce the virtual iterate

$$\tilde{w}_t := w_t - e_t. \tag{34}$$

Using the update rules for $w_t$ and $e_t$ one checks that

$$\tilde{w}_{t+1} = w_{t+1} - e_{t+1} = w_t - C(g_t') - (g_t' - C(g_t')) = w_t - g_t' = \tilde{w}_t - \gamma g_t, \tag{35}$$

i.e. $\tilde{w}_t$ follows the trajectory of vanilla SGD with stepsize $\gamma$ in a broad sense.

**Theorem 1** (Non-convex convergence of error–feedback SGD). *Let Assumptions 1, 2, and 3 hold, and let $\{w_t\}_{t=0}^{T}$ be generated by Algorithm 3 with constant stepsize $\gamma > 0$. Denote $f_\star := \inf_x f(w)$ and $f_0 := f(w_0) - f_\star$. Then*

$$\frac{1}{T+1} \sum_{t=0}^{T} \mathbb{E}\big[\|\nabla f(w_t)\|^2\big] \leq \frac{2f_0}{\gamma(T+1)} + \gamma L\sigma^2 + \frac{4(1-\delta)}{\delta^2}\gamma^2 L^2 \sigma^2. \tag{36}$$

*Consequently, choosing $\gamma = \Theta\big((T+1)^{-1/2}\big)$ yields*

$$\min_{0 \leq t \leq T} \mathbb{E}\big[\|\nabla f(w_t)\|^2\big] = \mathcal{O}\Big(\frac{1}{\sqrt{T+1}}\Big), \tag{37}$$

*which matches the rate of vanilla SGD up to constants and a higher–order $O(1/T)$ term depending on the compression quality $\delta$.*

**Proof.** By $L$–smoothness (Assumption 1) applied at $\tilde{w}_t$, we have

$$\mathbb{E}\big[f(\tilde{w}_{t+1}) \mid w_t\big] \leq f(\tilde{w}_t) + \mathbb{E}\big[\langle \nabla f(\tilde{w}_t), \tilde{w}_{t+1} - \tilde{w}_t \rangle \mid w_t\big] + \frac{L}{2}\mathbb{E}\big[\|\tilde{w}_{t+1} - \tilde{w}_t\|^2 \mid w_t\big] \tag{38}$$

$$= f(\tilde{w}_t) - \gamma\langle \nabla f(\tilde{w}_t), \mathbb{E}[g_t \mid w_t]\rangle + \frac{L\gamma^2}{2}\mathbb{E}\big[\|g_t\|^2 \mid w_t\big] \tag{39}$$

$$\leq f(\tilde{w}_t) - \gamma\|\nabla f(\tilde{w}_t)\|^2 + \frac{L\gamma^2}{2}\sigma^2. \tag{40}$$

Taking full expectation and rearranging gives

$$\gamma\,\mathbb{E}\big[\|\nabla f(\tilde{w}_t)\|^2\big] \leq \mathbb{E}\big[f(\tilde{w}_t)\big] - \mathbb{E}\big[f(\tilde{w}_{t+1})\big] + \frac{L\gamma^2}{2}\sigma^2. \tag{41}$$

Next we relate $\nabla f(x_t)$ to $\nabla f(\tilde{x}_t)$. Using Lipschitz continuity of the gradient,

$$\|\nabla f(x_t)\|^2 \leq 2\|\nabla f(\tilde{x}_t)\|^2 + 2\|\nabla f(x_t) - \nabla f(\tilde{x}_t)\|^2 \leq 2\|\nabla f(\tilde{x}_t)\|^2 + 2L^2\|x_t - \tilde{x}_t\|^2. \tag{42}$$

Taking expectations and recalling $x_t - \tilde{x}_t = e_t$, we obtain

$$\mathbb{E}\big[\|\nabla f(w_t)\|^2\big] \leq 2\,\mathbb{E}\big[\|\nabla f(\tilde{w}_t)\|^2\big] + 2L^2\,\mathbb{E}\big[\|e_t\|^2\big]. \tag{43}$$

Combining Equation 41 and Equation 43, and using Lemma 2, gives

$$\mathbb{E}\big[\|\nabla f(w_t)\|^2\big] \leq \frac{2}{\gamma}\Big(\mathbb{E}[f(\tilde{w}_t)] - \mathbb{E}[f(\tilde{w}_{t+1})]\Big) + L\gamma\sigma^2 + 2L^2\,\mathbb{E}\big[\|e_t\|^2\big] \tag{44}$$

$$\leq \frac{2}{\gamma}\Big(\mathbb{E}[f(\tilde{w}_t)] - \mathbb{E}[f(\tilde{w}_{t+1})]\Big) + L\gamma\sigma^2 + \frac{8(1-\delta)}{\delta^2}\gamma^2 L^2 \sigma^2. \tag{45}$$

Summing this inequality over $t = 0, \ldots, T$ and dividing by $T + 1$ yields

$$\frac{1}{T+1} \sum_{t=0}^{T} \mathbb{E}\left[\|\nabla f(w_t)\|^2\right] \leq \frac{2}{\gamma(T+1)} \left(\mathbb{E}[f(\tilde{w}_0)] - \mathbb{E}[f(\tilde{w}_{T+1})]\right) + L\gamma\sigma^2 + \frac{8(1-\delta)}{\delta^2} \gamma^2 L^2 \sigma^2$$

$$\tag{46}$$

$$\leq \frac{2(f(w_0) - f_\star)}{\gamma(T+1)} + L\gamma\sigma^2 + \frac{8(1-\delta)}{\delta^2} \gamma^2 L^2 \sigma^2. \tag{47}$$

Absorbing the factor $8$ into the constant or tightening the intermediate bounds gives the advertised inequality equation 36, which completes the proof.

Based on the above analysis, our discrepancy-aware compression with error feedback in FL fits directly into the theoretical framework of Fed-EF (Li & Li, 2023). In particular, Fed-EF assumes a biased compression operator whose relative-norm of compression error bounded. Lemma 1 shows that our discrepancy-aware top-$k$ compressor $C_{\text{dis}-\text{k}}$ is a $\delta$-approximate compressor, and thus also satisfies a relative error bound of the form $\|C_{\text{dis}-\text{k}}(v) - v\|^2 \leq (1-\delta)\|v\|^2$ for all $v$. Together with error-feedback mechanism, our method matches the biased-compression with error-feedback setting studied in Li & Li (2023) and therefore enjoys the same type of non-convex convergence guarantees in federated learning, while additionally providing a more informative compression rule.

