# OpenReview forum: "Enhancing Communication Compression via Discrepancy-aware Calibration for Federated Learning"
_ICLR.cc/2026/Conference — ICLR 2026 Poster_

### Official Review · Reviewer_qW7h · 2025-10-20

**Soundness:** 3
**Presentation:** 3
**Contribution:** 3
**Rating:** 6
**Confidence:** 4

**Summary:**

This work proposes a new way to do compression in distributed optimization, especially for federated learning. To that aim, the clients use a small subset of their local data as calibration data to measure the output discrepancy induced by compression.

**Strengths:**

Reducing the communication cost in distributed optimization, in particular for federated learning, is crucial. The approach of taking into account the discrepancy induced by compression is interesting. The experiments are convincing.

**Weaknesses:**

My main concern is that you use the word "heuristic" multiple times to characterize existing compression methods, in a dismissive and incorrect way. In fact, this is quite the opposite: existing approaches come with thorough theoretical guarantees, which is the contrary of being heuristic, but these guarantees are based on worst-case analysis, which is often too conservative. For instance, top-k and rand-k have the same guarantees, because the worst case vector to which they are applied is a vector will all elements equal, and in that case top-k and rand-k are equivalent. But top-k performs better in practice, and this is intuitive: it is better to keep the coordinates with largest amplitude, which capture the most information, than random coordinates chosen blindly. You should rephrase your statements and cite some papers, such as Beznosikov et al. "On biased compression for distributed learning," 2023; Condat et al. "EF-BV: A Unified Theory of Error Feedback and Variance Reduction Mechanisms for Biased and Unbiased Compression in Distributed Optimization," 2022; Yi et al. "FedComLoc: Communication-Efficient Distributed Training of Sparse and Quantized Models," 2025.

**Questions:**

No

---

> ### Author Response · Authors · 2025-11-19
> **Author Response**
>
> We sincerely thank the reviewer for the valuable feedback. We **have adjusted the wording** to avoid ambiguity, and explicitly **cited prior FL studies** on the convergence of compressed federated optimization.
>
> **(1) The “heuristic” we use does not refer to the lack of solid theoretical foundations in existing methods**
>
> What we meant by “heuristic” was specifically **the selection rules** commonly used in practice—such as keeping the largest-magnitude coordinates (Top-k), selecting random coordinates (rand-k), or truncating the smallest singular values in low-rank methods. These criteria are simple and practical, but they typically **do not** incorporate loss-, gradient-, or discrepancy-based information, and thus do not explicitly optimize the activation-dependent output error induced by dropping particular compressed units.
>
> At the same time, we **fully acknowledge the reviewer’s point** that compressors such as Top-k and rand-k come with rigorous worst-case theoretical guarantees, as established in prior works. We have now explicitly cited and discussed these results in the revised manuscript. Our contribution is complementary: rather than questioning the validity of these compressors, we focus on designing a selection mechanism that is explicitly guided by output discrepancy, which is not directly captured by the existing works.
>
> **(2) We have changed the wording to no longer use 'heuristic'**
>
> To avoid ambiguity, we **have revised the wording** in the revised manuscript. In particular, we have replaced the original phrase “heuristic” with the more precise description: “selection rules that do not incorporate loss- or discrepancy-related information.” This change makes it clear that our focus is the choice of selection criteria.
>
> We thank the reviewer again for pointing out this issue, and hope the revision can address your concern.

---

> > ### Comment · Reviewer_qW7h · 2025-11-20
> >
> > Thank you for taking my comments into account. For the moment, I am keeping my score.

---

> > > ### Author Response · Authors · 2025-11-20
> > > **Thank you for your prompt response!**
> > >
> > > Thank you for your prompt response and feedback. If you have any further questions or suggestions, either now or in the future, we would be happy to discuss them with you and make any necessary improvements. Wishing you a great day!

---

> > > ### Author Response · Authors · 2025-11-27
> > > **Highlight of the Latest Updated Version and Further Author Response**
> > >
> > > We sincerely thank you again for the insightful comments. Following your suggestion, we **have updated** the introduction accordingly **(Lines 60–65)** in the latest revised version.
> > >
> > > The newly added passage now reads:
> > >
> > > > “While these rules are simple and efficient, they do not account for the impact of dropping specific units on the resulting output discrepancy, which may cause important units to be discarded.
> > > This observation is consistent with the well-known gap between Top-k and Random-k in practical behavior, even though they share nearly identical worst-case bounds and convergence guarantees[1,2,3,4,5,6]. The reason is that preserving the largest-magnitude coordinates tends to retain more informative updates. Our method builds on and extends this intuition, as further discussed in Section 3.2.”
> > >
> > > In addition, we **have included** additional experiments in both the experiments and appendix sections to **demonstrate the advantage of our method**.
> > >
> > > We appreciate the reviewer’s guidance, which has helped improve the clarity of our paper. We would be very grateful if the revisions address your concerns and contribute positively to your evaluation.
> > >
> > > **Reference:**
> > >
> > > [1] On Biased Compression for Distributed Learning. J. Mach. Learn. Res. 2023.
> > >
> > > [2] EF-BV: A Unified Theory of Error Feedback and Variance Reduction Mechanisms for Biased and Unbiased Compression in Distributed Optimization. NeurIPS 2022.
> > >
> > > [3] Analysis of Error Feedback in Federated Non-Convex Optimization with Biased Compression: Fast Convergence and Partial Participation. ICML 2023.
> > >
> > > [4] Error Compensated Distributed SGD Can Be Accelerated. NeurIPS 2021.
> > >
> > > [5] FedComLoc: Communication-Efficient Distributed Training of Sparse and quantized models. Trans. Mach. Learn. Res., 2025.
> > >
> > > [6] Error Feedback Fixes SignSGD and other Gradient Compression Schemes. ICML 2019.

---

### Official Review · Reviewer_yzYJ · 2025-10-28

**Soundness:** 3
**Presentation:** 2
**Contribution:** 2
**Rating:** 6
**Confidence:** 4

**Summary:**

This paper tackles the key issue of communication costs in federated learning. Existing methods like Top-k sparsification and low-rank decomposition such as ATOMO depend on magnitude heuristics. These approaches fail in federated learning. Clients send accumulated parameter updates over local steps, not single gradients. This breaks the link between magnitude and real importance. The paper introduces a discrepancy-aware compression strategy. It avoids magnitude as an importance measure. Clients use a small local data subset for calibration. This directly assesses output changes from dropping each compression unit, like an element or singular triplet. The method keeps units with the largest output impact. It uses the limited bandwidth more wisely. The experiments demonstrate its effectiveness by augmenting both Top-k and ATOMO , achieving significant accuracy improvements, especially under high compression ratios and non-IID data distributions.

**Strengths:**

1.The paper's primary strength is its core insight. It correctly diagnoses why existing heuristics fail in the FL paradigm (parameter updates vs. gradients) and proposes a solution that directly exploits a key characteristic of FL (infrequent communication allows for more local computation). This is a fundamental contribution.

2.The experimental results are comprehensive and convincing. The method shows consistent, significant gains over baselines across multiple models, datasets, and compression types (element-wise and low-rank). The gains are most pronounced in the most challenging regimes: high compression and high data heterogeneity.

3.The paper is written with clarity. The motivation, method, and results are all easy to follow.

**Weaknesses:**

1.The overhead analysis in Table 4 is transparent about the per-round computational cost, showing a non-trivial increase (e.g., ~12-18% total time increase per round in some cases). The authors argue this is an acceptable trade-off. This argument would be much stronger if supported by a "time-to-accuracy" plot. Given the large accuracy gains, it is highly likely that the discrepancy-aware method converges in fewer rounds, potentially leading to a faster total wall-clock time to reach a target accuracy, even with the higher per-round cost. This analysis is currently missing.

2.The paper mentions PowerSGD in the introduction and states the framework is compatible with it. However, the low-rank experiments are conducted only with ATOMO. Given that PowerSGD is a more practical and widely used low-rank approximation than ATOMO (which requires a full SVD), demonstrating the "plug-in" capability and gains on PowerSGD would have significantly strengthened the paper's practical claims.

3.The paper's experiments are rightly focused on comparing the augmented methods to their original magnitude-based versions. However, the related work section mentions other advanced compression techniques, such as adaptive quantization (e.g., FedFQ, FedAQ). While the proposed method addresses selection and not quantization, a comparison or discussion of how it performs against other SOTA communication-efficiency methods (not just magnitude-based ones) would help situate its contribution in the broader landscape.

**Questions:**

1.The core idea is to find a better importance metric than magnitude. Could this discrepancy-aware metric $L_{comp}(u)$ also be used to guide other compression types, such as quantization?  Could it be used to allocate more bits to elements or singular triplets that have a high output discrepancy score?

2.The paper claims compatibility with PowerSGD. Were there any specific technical challenges that prevented its inclusion in the experiments? Would the application of the discrepancy metric be as straightforward as it is for ATOMO, given that PowerSGD approximates the SVD using iterative methods?

3.There is a subject-verb agreement error in lines 723–724: "the content being compressed (parameter updates) carry little gradient." The subject "the content" is singular, but the verb "carry" is plural. It should be corrected to "carries little gradient" to agree with the singular noun "content. You may also  check similar constructions elsewhere in the paper.

---

> ### Author Response · Authors · 2025-11-20
> **Author Response (1)**
>
> **W1: The overhead analysis in Table 4 is transparent about the per-round computational cost, showing a non-trivial increase (e.g., ~12-18% total time increase per round in some cases). The authors argue this is an acceptable trade-off. This argument would be much stronger if supported by a "time-to-accuracy" plot. Given the large accuracy gains, it is highly likely that the discrepancy-aware method converges in fewer rounds, potentially leading to a faster total wall-clock time to reach a target accuracy, even with the higher per-round cost. This analysis is currently missing.**
>
> Thank you for the insightful comment. Following the suggestion, we **have added** a new paragraph in Section 4.2 Main Results in the revised version. This paragraph now includes a "time-to-accuracy" analysis, demonstrating **the number of communication rounds required to achieve a target accuracy**.
>
> Specifically, we conduct experiments on two representative configurations at various compression ratios. The target accuracy is defined as 60$\%$ or 80$\%$ of the final test accuracy achieved by the discrepancy-aware method for each specific configuration.
>
> Across all configurations and compression ratios, the discrepancy-aware method **consistently** reaches the same target accuracy in **fewer rounds** than the magnitude-based baseline. In particular, it performs very well under strong compression, with a speedup of up to **1.56×** in reaching the target accuracy at a compression ratio of 0.01 on the CIFAR-10 dataset using the ViT-tiny model. This analysis demonstrates that discrepancy-aware compression **not only improves final accuracy**, but also **accelerates convergence** in terms of wall-clock time under the same communication budget.
>
> The newly added Tables as follows:
>
> (1) Magnitude-based vs. discrepancy-aware Top-k in FMNIST (AlexNet). Values in parentheses indicate the speedup of discrepancy-aware methods over magnitude-based baselines.
> | **Target Accuracy** | **Method**               | **0.01** | **0.05** | **0.1** | **0.2** | **0.4** | **0.6** |
> |---------------------|--------------------------|----------|----------|---------|---------|---------|---------|
> | 60% of the final test accuracy | **Magnitude-based**       | 73       | 54       | 55      | 54      | 56      | 56      |
> |       | **Discrepancy-aware**    | 62(**1.18×**) | 46(**1.17×**) | 50(**1.10×**) | 54(1.00×) | 56(1.00×) | 53(**1.05×**) |
> | 80% of the final test accuracy | **Magnitude-based**       | 105      | 85       | 83      | 83      | 84      | 81      |
> |       | **Discrepancy-aware**    | 82(**1.28×**) | 72(**1.18×**) | 71(**1.17×**) | 83(1.00×) | 81(**1.03×**) | 82(0.98×) |
>
> (2) Magnitude-based vs. discrepancy-aware Top-k in CIFAR-10 (ViT-tiny). Values in parentheses indicate the speedup of discrepancy-aware methods over magnitude-based baselines.
> | **Target Accuracy** | **Method**               | **0.01** | **0.05** | **0.1** | **0.2** | **0.4** | **0.6** |
> |---------------------|--------------------------|----------|----------|---------|---------|---------|---------|
> | 60% of the final test accuracy | **Magnitude-based**       | 53       | 45       | 33      | 36      | 39      | 40      |
> |       | **Discrepancy-aware**    | 44(**1.20×**) | 31(**1.45×**) | 26(**1.27×**) | 34(1.06×) | 37(1.05×) | 39(**1.03×**) |
> | 80% of the final test accuracy | **Magnitude-based**       | 128      | 117       |120      | 99      | 116      | 89      |
> |       | **Discrepancy-aware**    | 82(**1.56×**) | 81(**1.44×**) | 83(**1.45×**) | 79(1.25×) | 79(**1.47×**) | 86(1.03×) |

---

> ### Author Response · Authors · 2025-11-20
> **Author Response (2)**
>
> **W2&Q2: The paper mentions PowerSGD in the introduction and states the framework is compatible with it. However, the low-rank experiments are conducted only with ATOMO. Given that PowerSGD is a more practical and widely used low-rank approximation than ATOMO (which requires a full SVD), demonstrating the "plug-in" capability and gains on PowerSGD would have significantly strengthened the paper's practical claims.**
>
> Our discrepancy-aware principle is compatible with PowerSGD. We will provide **theoretical analysis** and **specific implementation** below. After that, we explain why we used ATOMO as our low-rank baseline in the current experiments. In the revised version, we **have added** the following analysis in Appendix A.3.
>
> **(1) Theoretical derivation and compatibility with PowerSGD**
>
> Our discrepancy-aware selection mechanism **only assumes** that a model update can be decomposed into **a sum of rank-1 components**. It does **not rely on** having an exact SVD. Concretely, for a low-rank approximation from PowerSGD, suppose a layer update matrix $W$ is decomposed as $W \approx \sum_{t=1}^{r} a_t b_t^\top$. The effect of removing the $t$-th component on the layer output is $$Y = (W_0 + W) X \rightarrow Y' = (W_0 + W - a_t b_t^\top) X = Y - a_t b_t^\top X.$$
>
> Thus, the contribution of component t to the output discrepancy on a local calibration set X is
> $$L\_{\text{comp}}(t)=\| \Delta_t \|\_F^2=\bigl\| a\_t b\_t^\top X \bigr\|\_F^2=\bigl\| a\_t (b\_t^\top X) \bigr\|\_F^2.$$
>
> This is exactly the quantity our method uses to rank and select components under a fixed communication budget. In ATOMO, the factorization comes from an exact SVD, so $(𝑎_𝑡, 𝑏_𝑡)$ correspond to singular vectors and the metric reduces to a function of the singular value and the right singular vector evaluated on 𝑋. In PowerSGD, we instead have a low-rank approximation $𝑊 ≈ 𝑃 𝑄^\top$, obtained via iterative power methods. The $t$-th columns $𝑝_𝑡$  and $𝑞_𝑡$ of $𝑃$ and $𝑄$ define a rank-1 component $𝑝\_𝑡 𝑞\_𝑡^\top$. Plugging $(𝑎\_𝑡, 𝑏\_𝑡)=(𝑝\_𝑡, 𝑞\_𝑡)$ into the formula above gives
> $L\_{\text{comp}}(t)=\bigl\| p\_t q\_t^\top X \bigr\|\_F^2=\bigl\| p\_t (q\_t^\top X) \bigr\|\_F^2,t = 1, \dots, r.$
> So our discrepancy metric applies to PowerSGD in exactly the same way, independent of how $𝑃$ and $𝑄$ were obtained.
>
> **(2) Specific implementation on top of PowerSGD**
>
> Given a standard PowerSGD implementation for a weight matrix, our discrepancy-aware selection would proceed as follows in each communication round:
>
> 1. Run PowerSGD to obtain a rank-{𝑟_cand} approximation 𝑊≈𝑃 𝑄^T for each compressed layer, where {𝑟_cand} is a pre-specified candidate rank that is intentionally larger than the effective rank allowed by the communication budget.
>
> 2. On a small local calibration set 𝑋, compute $L\_{\text{comp}}(t)$.
>
> 3. Rank the {𝑟_cand} components {𝑝_𝑡 𝑞_𝑡^T} by 𝐿_comp(𝑡) and select the top components that fit within the communication budget.
>
> 4. Form truncated factors 𝑃′ and 𝑄′ by keeping only the selected columns, and communicate 𝑃′, 𝑄′ instead of the full 𝑃, 𝑄.
>
> This procedure **does not** modify the internal power-iteration scheme or the encoding format of PowerSGD; it only changes which rank-1 components are ultimately transmitted, based on their measured impact on the layer output.
>
> **(3) Why we used ATOMO as the low-rank baseline**
>
> There were **no conceptual obstacles** to integrating PowerSGD; the choice to focus on ATOMO was mainly pragmatic. ATOMO provides an exact SVD-based low-rank compressor, which is a natural and widely used representative of low-rank methods. Using ATOMO as the baseline allowed us to study our discrepancy-aware selection in a setting where the low-rank factors are “ideal” (true singular vectors). In this way, while demonstrating the performance improvement of our discrepancy-aware selection rule, other approximate SVD methods are theoretically compatible. Therefore, we prioritized implementing and analyzing the ATOMO-based variant as a clean, representative low-rank baseline.
>
> In the revised version, we **have added** the above analysis in Appendix A.3.

---

> ### Author Response · Authors · 2025-11-20
> **Author Response (3)**
>
> **W3: The paper's experiments are rightly focused on comparing the augmented methods to their original magnitude-based versions. However, the related work section mentions other advanced compression techniques, such as adaptive quantization (e.g., FedFQ, FedAQ). While the proposed method addresses selection and not quantization, a comparison or discussion of how it performs against other SOTA communication-efficiency methods (not just magnitude-based ones) would help situate its contribution in the broader landscape.**
>
> FedFQ and FedAQ are **quantization-based** methods, whereas our work **focuses on compression**. As discussed in the Section 2 Related Work,  compression and quantization are two mainstream approaches for communication-efficient Federated Learning[1], both aiming to mitigate limited communication bandwidth, but their core ideas are **fundamentally distinct**. Compression addresses **what to transmit**, while quantization focuses on **how to represent transmitted information with fewer bits**.
>
> Futhermore, compression and quantization are **orthogonal** and **can be combined** synergistically in FL pipelines. For instance: STC [2], FedQCS [3], and FedCSTQ [4] all combine compression with low-bit quantization.
>
> Therefore, our current experiments focus on Top-k and ATOMO as representative mainstream compression methods, which align with our method's design as a plug-in module to enhance selection rule in compression schemes.
>
> **References:**
>
> [1] A Comprehensive Survey on Communication-Efficient Federated Learning in Mobile Edge Environments, IEEE CST 2025
>
> [2] Robust and Communication-Efficient Federated Learning From Non-i.i.d. Data. IEEE TNNLS 2020
>
> [3] Communication-Efficient Federated Learning via Quantized Compressed Sensing. IEEE TWC 2023
>
> [4] Communication-efficient federated learning based on compressed sensing and ternary quantization. Appl. Intell. 2025
>
> ---
>
> **Q1: The core idea is to find a better importance metric than magnitude. Could this discrepancy-aware metric also be used to guide other compression types, such as quantization? Could it be used to allocate more bits to elements or singular triplets that have a high output discrepancy score?**
>
> Thank you for your insightful comment. We believe this line of thinking is indeed feasible and maybe a highly promising direction for future exploration. In the revised version, we **have added** the analysis in Appendix A.4. Our proposed discrepancy-aware principle is **not limited** to compression methods in FL, and that extending it to guide quantization and bit allocation is **a promising direction** for future research.
>
> In the initial design, our discrepancy-aware principle **is designed for** compression methods. It only requires the ability to simulate the effect of a given compression operation on the layer outputs over a small calibration set. In the paper, we instantiate this idea for element-wise sparsification (Top-k) and low-rank decomposition (ATOMO) by treating an element or a singular triplet as the compression unit and scoring it via the induced output discrepancy.
>
> This same principle **naturally extends to** quantization-based methods. Concretely, we can define the quantization unit $u$, and consider multiple candidate bit-widths b∈B for that unit. For each pair $(u,b)$, we can construct a version of the update where $u$ is quantized with $b$ bits while other units are kept at a reference precision. Then measure the corresponding discrepancy metric on the calibration data. Given a layer bit budget, one can then allocate bits by prioritizing units for which increasing the bit-width yields the largest reduction in the discrepancy metric per additional bit. This yields a discrepancy-aware quantization scheme, where high-discrepancy units receive more bits and low-discrepancy units can be more aggressively quantized.
>
> We view this as complementary to FL quantization frameworks. Our discrepancy-aware principle **can be plugged into** such frameworks. We believe that the experimental results of the specific design based on this idea will be promising, which may be an interesting topic in the field of quantification.
>
> In the revised version, we **have clarified** in Appendix A.4 that the proposed discrepancy-aware principle is not limited to compression methods in FL, and that extending it to guide quantization and bit allocation is a promising direction for future research.

---

> ### Author Response · Authors · 2025-11-20
> **Author Response (4)**
>
> **Q3: There is a subject-verb agreement error in lines 723–724: "the content being compressed (parameter updates) carry little gradient." The subject "the content" is singular, but the verb "carry" is plural. It should be corrected to "carries little gradient" to agree with the singular noun "content. You may also check similar constructions elsewhere in the paper.**
>
> We thank the reviewer for the careful reading and for pointing out the subject–verb agreement error at lines 723–724. We **have revised** "carry" to "carries" to agree with the singular noun "content". Meanwhile, we **have carefully proofread** the entire manuscript to identify and fix similar issues, thereby improving the readability of the revised version.

---

> > ### Comment · Reviewer_yzYJ · 2025-11-20
> >
> > The rebuttal has clarified the majority of my confusion. I appreciate the clarifications and will increase my score accordingly.

---

> > > ### Author Response · Authors · 2025-11-20
> > > **Thank you for your quick response!**
> > >
> > > Thank you for your quick response and positive feedback. We're glad to hear that the majority of your confusion have been clarified, and we truly appreciate you raising the score. Have a nice day!

---

### Official Review · Reviewer_atF7 · 2025-10-30

**Soundness:** 2
**Presentation:** 2
**Contribution:** 2
**Rating:** 4
**Confidence:** 3

**Summary:**

This paper proposes a discrepancy-aware compression method for Federated Learning. It uses a small local calibration dataset to measure the output impact of dropping compression units, replacing conventional magnitude-based rules. This approach serves as a plug-in to enhance existing methods like Top-k and ATOMO, significantly boosting accuracy under tight communication budgets.

**Strengths:**

1. Results showing significant accuracy gains under high compression ratios.
2. The method is designed as a plug-in, making it compatible with existing compression schemes.
3. Provides theoretical motivation and intuitive examples to illustrate the limitations of magnitude-based heuristics.

**Weaknesses:**

1.While the proposed method introduces non-negligible computational overhead due to the calibration process—a significant concern for resource-constrained devices—the empirical validation of its performance gain is not sufficiently comprehensive. The scope of tested compression methods is limited to Top-k and ATOMO. To fully justify the incurred overhead, it is crucial to demonstrate the method's effectiveness across a broader spectrum of techniques, such as SignSGD/z-SignSGD and other quantization-based methods.

2. Lack of comparison with FedAvg or other uncompressed baselines to illustrate the absolute performance gap.

3. No discussion on whether the method can be combined with compensation mechanisms (e.g., error feedback) to further reduce compression loss.

4.The experiments are confined to computer vision tasks. The effectiveness on NLP datasets and models remains unverified, limiting the generalizability of the claims.

**Questions:**

Please see weaknesses

---

> ### Author Response · Authors · 2025-11-19
> **Author Response (1)**
>
> Thank you for your insightful feedback.
>
> ---
>
> **W1: While the proposed method introduces non-negligible computational overhead due to the calibration process—a significant concern for resource-constrained devices—the empirical validation of its performance gain is not sufficiently comprehensive. The scope of tested compression methods is limited to Top-k and ATOMO. To fully justify the incurred overhead, it is crucial to demonstrate the method's effectiveness across a broader spectrum of techniques, such as SignSGD/z-SignSGD and other quantization-based methods.**
>
> We provide a detailed explanation below.
>
> **(1) Computational overhead vs. performance gain**
>
> Our method is specifically designed for **highly heterogeneous** FL under **strongly constrained communication**, where communication is the primary bottleneck. In such regimes, it is standard in FL to spend additional local computation to better utilize bandwidth. Conversely, when communication is abundant to transmit almost all updates, different selection rules yield almost identical subsets, as shown in **Figure 1**, so the choice of rule becomes largely irrelevant.
>
> Our method provides **substantial communication-efficiency gains**: as shown in Tables 1 and 2 and the added Table 3 and 4, discrepancy-aware compression consistently **improves final accuracy** under high compression ratios and can reach a target accuracy with **only ~64% of the communication rounds** required by magnitude-based methods. In return, as reported in Table 4,  discrepancy-aware compression increases the per-round client time only modestly (e.g., from 0.569s to 0.638s for Top-k with ViT-tinyl). This overhead is acceptable in FL, where communication is infrequent and uplink bandwidth dominates the cost.
>
> Therefore, in the intended regime of strongly constrained communication and heterogeneous clients, the modest extra computation is well justified by the empirical performance and communication savings.
>
> **(2) Why focus on Top-k and ATOMO as baselines?**
>
> Our goal is to study selection rules for communication compression, rather than to compete with FL systems that stacks multiple orthogonal modules. Top-k and ATOMO are canonical representatives of the two dominant compression families.
>
> As far as we know, follow-up works [1,2,3,4,6] in these families mainly integrate it with other techniques (e.g., privacy and coding schemes), rather than fundamentally changing the selection rule itself. Evaluating our discrepancy-aware selection on these two canonical baselines thus allows us to:
>
> **1. Isolate and fairly evaluate the contribution of the selection rule.** If we compared against methods with multiple additional modules, it would become difficult to attribute gains to the selection rule versus other components.
>
> **2. Demonstrate broad applicability.** By showing consistent gains for both Top-k and ATOMO across multiple datasets and architectures (Tables 1 and 2), we provide evidence that our discrepancy-aware selection is a general plug-in rather than a method tuned to a specific compressor.
>
> Due to space and complexity constraints, we do not exhaustively evaluate every derivative of Top-k/ATOMO. However, since those methods keep the same magnitude-based selection core, In principle, our strategy can also be directly implemented as a plugin to enhance communication efficiency under compression.
>
> **(3) Regarding SignSGD/z-SignSGD and other quantization-based methods**
>
> SignSGD and z-SignSGD are quantization-based methods, whereas our work focuses on compression. As discussed in Section 2,  compression and quantization are two mainstream approaches for communication-efficient FL[5]. Both mitigate limited bandwidth, but their core ideas differ. Compression addresses what to transmit, while quantization addresses how to represent transmitted information with fewer bits.
>
> Futhermore, compression and quantization are orthogonal and can be combined. For instance: STC[6], FedQCS[7], and FedCSTQ[8] all combine compression with low-bit quantization.
>
> **Reference:**
>
> [1] Sparse Communication for Distributed Gradient Descent. EMNLP 2017
>
> [2] Deep Gradient Compression: Reducing the Communication Bandwidth for Distributed Training. ICLR 2018
>
> [3] Federated Learning with Sparsified Model Perturbation: Improving Accuracy under Client-Level Differential Privacy. IEEE TMC 2024
>
> [4] PowerSGD: Practical Low-Rank Gradient Compression for Distributed Optimization. NeurIPS 2019
>
> [5] A Comprehensive Survey on Communication-Efficient Federated Learning in Mobile Edge Environments, IEEE CST 2025
>
> [6] Robust and Communication-Efficient Federated Learning From Non-i.i.d. Data. IEEE TNNLS 2020
>
> [7] Communication-Efficient Federated Learning via Quantized Compressed Sensing. IEEE TWC 2023
>
> [8] Communication-efficient federated learning based on compressed sensing and ternary quantization. Appl. Intell. 2025

---

> ### Author Response · Authors · 2025-11-19
> **Author Response (2)**
>
> **W2: Lack of comparison with FedAvg or other uncompressed baselines to illustrate the absolute performance gap.**
>
> To illustrate the absolute performance gap, we have included results with **a compression ratio of 1.0** (i.e., **Fedavg without compression**) in the revised Table 1. The previous experimental results have already demonstrated the effectiveness of our method, and the uncompressed FedAvg baseline will further highlight the performance gains.
>
> **Revised Table 1** is as following:
>
> Table 1: Final test accuracy of the element-wise sparsification method Top-k and the corresponding
> discrepancy-aware augmented method across different compression ratios, datasets, and models.
> | Dataset (Model)      | Method            | 0.01              | 0.1               | 0.2               | 0.4               | 0.6               | 1.0               |
> |----------------------|------------------|-------------------|-------------------|-------------------|-------------------|-------------------|-------------------|
> | CIFAR-10 (ViT-tiny)  | Magnitude-based  | 21.03 ±0.3        | 34.93 ±0.5        | 37.69 ±0.3        | 38.25 ±0.2        | 38.57 ±0.2        | 51.56 ±0.3    |
> |                      | Discrepancy-aware| **29.62 ±0.1**    | **41.52 ±0.4**    | **43.11 ±0.4**    | **42.81 ±0.1**    | **41.21 ±0.2**    |                    51.56 ±0.3    |
> | CIFAR-100 (ViT-small)| Magnitude-based  | 9.29 ±0.4         | 26.91 ±0.3        | 28.51 ±0.5        | 29.97 ±0.4        | 30.23 ±0.5        | 34.13 ±0.2    |
> |                      | Discrepancy-aware| **13.29 ±0.2**    | **28.62 ±0.3**    | **31.27 ±0.1**    | **32.87 ±0.4**    | **33.32 ±0.3**    |                    34.13 ±0.2    |
> | CIFAR-100 (ResNet-18)| Magnitude-based  | 10.76 ±0.1        | 27.28 ±0.7        | 30.71 ±1.0        | 32.34 ±0.9        | 32.82 ±0.6        | 35.33 ±0.1    |
> |                      | Discrepancy-aware| **15.58 ±0.3**    | **29.71 ±0.4**    | **32.54 ±0.5**    | **33.65 ±0.7**    | **33.71 ±0.4**    |                    35.33 ±0.1    |
> | Fashion-MNIST (AlexNet)| Magnitude-based| 63.31 ±0.1        | 70.32 ±0.9        | 71.50 ±0.5        | 71.59 ±0.7        | 71.98 ±0.7        | 78.96 ±0.3    |
> |                      | Discrepancy-aware| **67.55 ±0.3**    | **73.42 ±0.7**    | **73.61 ±0.4**    | **73.70 ±0.4**    | **74.01 ±0.5**    |                   78.96 ±0.3    |
>
> ---
>
> **W3: No discussion on whether the method can be combined with compensation mechanisms (e.g., error feedback) to further reduce compression loss.**
>
> Our method is **fully compatible** with compensation mechanisms. In fact, in all originally reported experiments, the standard error-feedback (EF) scheme was **already included** in the experimental configuration. In the revised version, we **have added related explanations** in the method and experiment sections.
>
> Specifically, each client maintains a residual $e_t=w_t+e_{t-1}-C(w_t+e_{t-1})$ and transmit a compressed update $C(w_t+e_{t-1})$, i.e., the compression error from the previous round is added back before compression in the current round. This is the classic EF scheme, which is known to mitigate the bias of compressed updates and markedly reduce compression loss.
>
> To avoid ambiguity, in the revised version, we have (i) explicitly stated in the experimental section that **EF is used for all results**, and (ii) explicitly stated **how EF is integrated into our pipeline** in the methods section.

---

> ### Author Response · Authors · 2025-11-21
> **Author Response (3)**
>
> **W4: The experiments are confined to computer vision tasks. The effectiveness on NLP datasets and models remains unverified, limiting the generalizability of the claims.**
>
> We sincerely thank the reviewer for the constructive suggestion. To verify the generalizability of our method beyond CV tasks, we **conducted additional experiments on four NLP datasets** (20News, AG News, CoLA, and MNLI) using DistilBERT.
>
> **Experimental Setup:**
> We used a consistent Non-IID setting ($\alpha=0.2$) with 100 total clients (10 participants per round). Each client trained for 2 local epochs ($E=2$) with a learning rate of $2\times10^{-5}$ over 200 global rounds. These configurations are mostly consistent with the CV tasks in Section 4, with only the learning rate adjusted to accommodate NLP tasks and specific model.
>
> **NLP Datasets:**
> We evaluate our method on four standard NLP benchmarks: 20News, AG News, CoLA, and MNLI.
> **20News [1]**. The 20 Newsgroups (20News) dataset is a collection of roughly 20,000 Usenet posts evenly distributed across 20 different newsgroups, and is a classic benchmark for topic-based text classification.
> **AG News [2]**. AG News is a news topic classification dataset constructed from AG’s corpus by selecting four top-level categories (World, Sports, Business, Sci/Tech), with 120K training and 7.6K test samples.
> **CoLA [3]**. The Corpus of Linguistic Acceptability (CoLA) is a binary sentence-level classification dataset, where each sentence is labeled as grammatically acceptable or unacceptable.
> **MNLI [4]**. The Multi-Genre Natural Language Inference (MNLI) corpus is a large-scale natural language inference benchmark containing about 433K premise–hypothesis sentence pairs from ten distinct genres of written and spoken English, annotated with three labels (entailment, contradiction, neutral).
>
> **Results:**
> The results demonstrate that our method **consistently outperforms** the magnitude-based baseline in both **accuracy** (NLP-Tasks Table 1 and **convergence speed** (NLP-Tasks Table 2) under extreme compression scenarios ($0.05\% - 1\%$). In the revised version, these results and detialed results will also be concluded in the Appendix A.6.
>
> **NLP-Tasks Table 1: Final Accuracy Comparison on NLP Datasets**
>
> | Dataset | Method | 0.0005 | 0.001 | 0.005 | 0.01 | 1.0 (Base) |
> | :--- | :--- | :--- | :--- | :--- | :--- | :--- |
> | **20News** | Magnitude | 10.20 ±0.9 | 14.84 ±0.8 | 44.26 ±0.6 | 50.45 ±0.5 | 66.25 |
> | | **LossAware**| **24.30 ±0.7**| **31.65 ±0.6**| **45.57 ±0.4**| **58.56 ±0.4**| 66.25 |
> | **AG News**| Magnitude | 70.80 ±1.1 | 86.77 ±0.6 | 89.01 ±0.4 | **90.25 ±0.2**| 90.88 |
> | | **LossAware**| **80.98 ±0.6**| **88.95 ±0.4**| **90.04 ±0.2**| 90.03 ±0.2 | 90.88 |
> | **CoLA** | Magnitude | 69.10 ±1.2 | 69.26 ±0.8 | 69.26 ±0.6 | 68.80 ±0.5 | 71.35 |
> | | **LossAware**| **69.22 ±0.8**| **69.26 ±0.6**| **69.26 ±0.4**| **69.45 ±0.4**| 71.35 |
> | **MNLI** | Magnitude | 38.08 ±1.5 | 55.45 ±0.9 | 67.34 ±0.5 | 70.81 ±0.3 | 71.96 |
> | | **LossAware**| **62.09 ±0.8**| **64.26 ±0.6**| **68.90 ±0.4**| **70.89 ±0.2**| 71.96 |
>
> **NLP-Tasks Table 2: Communicationr ounds required to reach target accuracy in NLP tasks. Values in parentheses indicate the speedup. "NR" denotes Target Not Reached.**
>
> | Dataset | Target Acc | Method | 0.0005 | 0.001 | 0.005 | 0.01 |
> | :--- | :--- | :--- | :--- | :--- | :--- | :--- |
> | **20News** | 60% Final | Magnitude | NR | NR | 108 | 69 |
> | | | **LossAware** | **57 (-)** | **68 (-)** | **46 (2.35×)**| **52 (1.33×)**|
> | | 80% Final | Magnitude | NR | NR | 134 | 100 |
> | | | **LossAware** | **126 (-)** | **131 (-)**| **82 (1.63×)**| **95 (1.05×)**|
> | **AG News**| 60% Final | Magnitude | 51 | 20 | 16 | 7 |
> | | | **LossAware** | **20 (2.55×)**| **10 (2.00×)**| **10 (1.60×)**| **3 (2.33×)** |
> | | 80% Final | Magnitude | 90 | 62 | 20 | 16 |
> | | | **LossAware** | **32 (2.81×)**| **17 (3.65×)**| **18 (1.11×)**| **15 (1.07×)**|
> | **CoLA** | 60% Final | Magnitude | 12 | 12 | 12 | 12 |
> | | | **LossAware** | **9 (1.33×)** | **9 (1.33×)** | **9 (1.33×)** | **9 (1.33×)** |
> | | 80% Final | Magnitude | 43 | 25 | 16 | 15 |
> | | | **LossAware** | **25 (1.72×)**| **22 (1.14×)**| **14 (1.14×)**| **14 (1.07×)**|
> | **MNLI** | 60% Final | Magnitude | 44 | 21 | 11 | 11 |
> | | | **LossAware** | **15 (2.93×)**| **13 (1.62×)**| 11 (1.00×) | **10 (1.10×)**|
> | | 80% Final | Magnitude | 152 | 80 | 49 | 40 |
> | | | **LossAware** | **69 (2.20×)**| **49 (1.63×)**| **34 (1.44×)**| **31 (1.29×)**|
>
> **Reference:**
>
> [1]  Newsweeder: Learning to filter netnews. ICML 1995.
>
> [2] Character-level Convolutional Networks for Text Classification. NeurIPS 2015.
>
> [3] Neural Network Acceptability Judgments. Trans. Assoc. Comput. Linguistics 7: 625-641 (2019).
>
> [4] A Broad-Coverage Challenge Corpus for Sentence Understanding through Inference. NAACL-HLT 2018

---

> ### Author Response · Authors · 2025-11-25
> **Looking forward to your feedback!**
>
> Dear Reviewer atF7,
>
> I hope this message finds you well. Thank you once again for your valuable feedback. We have conducted additional experiments and made revisions to the paper based on your suggestions. As the discussion phase is nearing its end, we would like to know if our responses have addressed your concerns. We look forward to hearing from you.
>
> Thank you for your time and effort in reviewing our paper.
>
> Best regards,
>
> Authors

---

### Official Review · Reviewer_qMu3 · 2025-10-31

**Soundness:** 2
**Presentation:** 3
**Contribution:** 2
**Rating:** 4
**Confidence:** 4

**Summary:**

Federated Learning (FL) enables privacy-preserving collaborative training by allowing clients to jointly learn a shared model without exchanging raw data. However, the communication overhead from model updates remains a key bottleneck, especially for resource-constrained devices. Existing compression methods largely rely on magnitude-based (e.g., Top-k) or randomness-based heuristics (e.g., ATOMO, PowerSGD), which ignore the discrepancy between compressed and original outputs, often leading to critical information loss. They propose a discrepancy-aware compression method that significantly boosts performance under extreme communication constraints. Each client uses a small subset of its local data for calibration to directly measure the output discrepancy caused by dropping candidate compression units and uses this as a selection criterion. This strategy can be integrated into mainstream compression schemes to enhance communication efficiency. Experiments show a 18.9% relative accuracy improvement at a compression ratio of 0.1, demonstrating its effectiveness for scalable and communication-efficient FL.

**Strengths:**

1. The paper addresses an important problem with insightful observation.
2. The paper is well-written and clearly organized.

**Weaknesses:**

1.	Traditional compression algorithms come with theoretical convergence guarantees. Does this algorithm also ensure convergence?
2.	The experiments suffer from significant shortcomings. In fact, the efficiency of conventional distributed communication compression methods largely stems from their use of error compensation mechanisms, yet this paper does not compare against such approaches in its baselines.
3.	In traditional distributed communication settings, compression ratios are typically above 1%; in contrast, the highest compression ratio evaluated in this paper is only 10%.

**Questions:**

Please refer to weaknesses.

---

> ### Author Response · Authors · 2025-11-19
> **Author Response (1)**
>
> **W1: Traditional compression algorithms come with theoretical convergence guarantees. Does this algorithm also ensure convergence?**
>
> Thank you for the constructive comments. Following your suggestions, we **have added a detailed convergence analysis** in Appendix A.5.
>
> **(1) From the theoretical perspective**
>
> Our Discrepancy-aware compressor shares the same type of **convergence guarantees** as the classical top-$k$ compressor with error–feedback.
> Following previous works on compressor convergence in distributed learning [1, 2, 3, 4, 5], we adopted similar approaches to analyze the convergence of our discrepancy-aware compressor.
>
> We formally prove that our discrepancy-aware compressor with error–feedback has the same type of non-convex convergence guarantees as traditional compression schemes. First, Lemma 1 shows that the discrepancy-aware top-$k$ operator $C_{\mathrm{dis\text{-}k}}$ is a $\delta$-approximate compressor on the feasible set $Q$.
> Building on this, Lemma 2 indicates that, under standard assumptions of $L$–smoothness and unbiased stochastic gradients with bounded variance, the error–feedback recursion keeps **the compression error uniformly bounded**:
> $
> \mathbb{E}\big[\|e_t\|^2\big]
> \\;\le\\;
> \frac{4(1-\delta)}{\delta^2}\,\gamma^2 \sigma^2,
> \qquad \forall t \ge 0.
> $
>
> By introducing the virtual iterate $\tilde{w}\_t = w\_t - e\_t$, we show that $\tilde{w}\_t$ follows the same update as vanilla SGD, which allows us to derive **a standard non-convex convergence bound**. Specifically, Theorem 1 proves that, for a constant stepsize $\gamma > 0$, choosing $\gamma = \Theta\big((T+1)^{-1/2}\big)$ yields $\min_{0\le t\le T} \mathbb{E}\big[\|\nabla f(w_t)\|^2\big] = \mathcal{O}\Big(\frac{1}{\sqrt{T+1}}\Big),$
> which matches the convergence rate of vanilla SGD up to constants depending on $\delta$ and a higher–order $\mathcal{O}(1/T)$ term.
>
> Finally, we clarify that this analysis aligns our method with the Fed-EF framework [6]: our discrepancy-aware compressor satisfies the same **relative error condition** $\|C_{\mathrm{dis\text{-}k}}(v)-v\|^2 \le (1-\delta)\|v\|^2$ as the biased compressors studied there. Together with the error–feedback mechanism, our federated learning algorithm therefore has rigorous **non-convex convergence guarantees** similar to those of traditional compression-based methods, while providing a more informative compression rule.
>
> **(2) From the empirical perspective**
>
> Our Discrepancy-aware compressor exhibits **superior performance** in federated learning scenarios with high data heterogeneity and stringent communication budgets.
> In the revised version, **Tables 1 and 2** demonstrate that our discrepancy-aware compressor consistently achieves **higher test accuracy** than the baselines across all considered datasets and heterogeneity levels. Moreover, **the new added Tables 3 and 4** show that our approach can reach the same target accuracy with **substantially fewer communication rounds**. This indicates that our method not only **converges faster in terms of communication efficiency**, but also has **better or comparable final accuracy**, highlighting clear advantages in both convergence speed and ultimate performance under practical FL constraints.
>
> **References:**
>
> [1] Error Feedback Fixes SignSGD and other Gradient Compression Schemes. ICML 2019.
>
> [2] On Biased Compression for Distributed Learning. J. Mach. Learn. Res. 2023.
>
> [3] EF-BV: A Unified Theory of Error Feedback and Variance Reduction Mechanisms for Biased and Unbiased Compression in Distributed Optimization. NeurIPS 2022
>
> [4] On the Bias-Variance-Cost Tradeoff of Stochastic Optimization. NeurIPS 2021.
>
> [5] Error Compensated Distributed SGD Can Be Accelerated. NeurIPS 2021
>
> [6] Analysis of Error Feedback in Federated Non-Convex Optimization with Biased Compression: Fast Convergence and Partial Participation. ICML 2023.

---

> ### Author Response · Authors · 2025-11-19
> **Author Response (2)**
>
> **W2: The experiments suffer from significant shortcomings. In fact, the efficiency of conventional distributed communication compression methods largely stems from their use of error compensation mechanisms, yet this paper does not compare against such approaches in its baselines.**
>
> In fact, **in all originally reported experiments**, the standard **error-feedback (EF)** scheme **was already included** in the experimental configuration. **Baseline and our compressor** used the **same** error compensation mechanism in the experimental comparison to ensure fairness. In the revised version, we **have added** relevant explanations in the method and experiment sections.
>
> Specifically, each client maintains a residual $e\_t=w\_t+e\_{t-1}-C(w\_t+e\_{t-1})$ and transmit a compressed update $C(w\_t+e\_{t-1})$, i.e., the compression error from the previous round is added back before compression in the current round. This is the classic EF scheme, which is known to mitigate the bias of compressed updates and markedly reduce compression loss.
>
> To avoid ambiguity, in the revised version, we have (i) explicitly stated in the experimental section that **EF is used for all results**, and (ii) explicitly stated **how EF is integrated into our pipeline** in the methods section.
>
> ---
>
> **W3: In traditional distributed communication settings, compression ratios are typically above 1%; in contrast, the highest compression ratio evaluated in this paper is only 10%.**
>
> Thank you for the constructive comments.
> In the revised version, we **have added** experimental results for a compression ratio of **0.01** in the revised Table 1, where our method still achieves better performance than baselines.
> For example, on the CIFAR-10 dataset with the ViT-Tiny model, the traditional Top-k method **exhibits severe oscillations** in the loss curve at a compression ratio of 0.01, which is very difficult to converge. In contrast, our method is able to **maintain stable convergence** under the same setting, which leads to **a substantial gap in the final test accuracy**. After incorporating these new experimental results, the range of compression configurations in our experiments is now broadly aligned with the configurations used in other non-IID federated learning (FL) settings[1, 2, 3].
>
> **Revised Table 1** is as following:
>
> Table 1: Final test accuracy of the element-wise sparsification method Top-k and the corresponding
> discrepancy-aware augmented method across different compression ratios, datasets, and models.
> | Dataset (Model)      | Method            | 0.01              | 0.1               | 0.2               | 0.4               | 0.6               | 1.0               |
> |----------------------|------------------|-------------------|-------------------|-------------------|-------------------|-------------------|-------------------|
> | CIFAR-10 (ViT-tiny)  | Magnitude-based  | 21.03 ±0.3        | 34.93 ±0.5        | 37.69 ±0.3        | 38.25 ±0.2        | 38.57 ±0.2        | 51.56 ±0.3    |
> |                      | Discrepancy-aware| **29.62 ±0.1**    | **41.52 ±0.4**    | **43.11 ±0.4**    | **42.81 ±0.1**    | **41.21 ±0.2**    |                    51.56 ±0.3    |
> | CIFAR-100 (ViT-small)| Magnitude-based  | 9.29 ±0.4         | 26.91 ±0.3        | 28.51 ±0.5        | 29.97 ±0.4        | 30.23 ±0.5        | 34.13 ±0.2    |
> |                      | Discrepancy-aware| **13.29 ±0.2**    | **28.62 ±0.3**    | **31.27 ±0.1**    | **32.87 ±0.4**    | **33.32 ±0.3**    |                    34.13 ±0.2    |
> | CIFAR-100 (ResNet-18)| Magnitude-based  | 10.76 ±0.1        | 27.28 ±0.7        | 30.71 ±1.0        | 32.34 ±0.9        | 32.82 ±0.6        | 35.33 ±0.1    |
> |                      | Discrepancy-aware| **15.58 ±0.3**    | **29.71 ±0.4**    | **32.54 ±0.5**    | **33.65 ±0.7**    | **33.71 ±0.4**    |                    35.33 ±0.1    |
> | Fashion-MNIST (AlexNet)| Magnitude-based| 63.31 ±0.1        | 70.32 ±0.9        | 71.50 ±0.5        | 71.59 ±0.7        | 71.98 ±0.7        | 78.96 ±0.3    |
> |                      | Discrepancy-aware| **67.55 ±0.3**    | **73.42 ±0.7**    | **73.61 ±0.4**    | **73.70 ±0.4**    | **74.01 ±0.5**    |                   78.96 ±0.3    |
>
>
> **References:**
>
> [1] CFedAvg: Achieving Eﬀicient Communication and Fast Convergence in Non-IID Federated Learning. 2021
>
> [2] FedFQ: Federated Learning with Fine‑Grained Quantization. 2024.
>
> [3] CriticalFL: A Critical Learning Periods Augmented Client Selection Framework for Efficient Federated Learning. KDD 2023

---

> ### Author Response · Authors · 2025-11-25
> **Looking forward to your feedback!**
>
> Dear Reviewer qMu3,
>
> I hope this message finds you well. Thank you once again for your valuable feedback. We have conducted additional experiments and made revisions to the paper based on your suggestions. As the discussion phase is nearing its end, we would like to know if our responses have addressed your concerns. We look forward to hearing from you.
>
> Thank you for your time and effort in reviewing our paper.
>
> Best regards,
>
> Authors

---

### Author Response · Authors · 2025-12-01
**Summary of Rebuttal**

Dear **AC,**

Thank you very much for your efforts for the community. Since the discussion phase has ended, we would like to briefly **summarize our rebuttal** to assist your decision.

This paper proposes a **discrepancy-aware compression** method for Federated Learning. It uses a small local calibration dataset to measure the output impact of dropping compression units, replacing conventional magnitude-based selection rules. This approach serves as a **plug-in** to enhance existing methods like Top-k and ATOMO, significantly **boosting accuracy and convergence speed** under tight communication budgets.

During the discussion, we carefully addressed each concern raised by the reviewers, providing detailed explanations and making corresponding modifications. After the rebuttal, **Reviewer yzYJ has raised the score from (6) to (8)** before the OpenReview information leak issues.

The specific modifications are as follows:

- We **added a detailed convergence analysis in Appendix A.5**, proving that our discrepancy-aware compressor with error-feedback has non-convex convergence guarantees similar to traditional compression schemes, aligning with previous work on distributed learning compressors. **[@Reviewer_qMu3 W1]**

- We **clarified** the inclusion of the error-feedback (EF) scheme in all original experiments and **stated** how EF is integrated into our pipeline in the revised version. **[@Reviewer_qMu3 W2]** **[@Reviewer_atF7 W3]**

- We **included new experimental results** in Table 1, showing the performance of our method at very low compression ratios (0.01). The results demonstrate that our approach outperforms traditional Top-k methods in terms of stability and final test accuracy. **[@Reviewer_qMu3 W3]**

- We **addressed concerns** about the computational overhead by demonstrating, through **revised Table 1, 2, 3, and 4**, that the modest extra computationis justified by significant improvements in both accuracy and convergence speed. Additionally, we **explained** why the experimental configurations used in the paper are appropriate. By focusing on foundational methods like Top-k and ATOMO, we demonstrate the **importance of improving the selection rule**. This not only showcases the **effectiveness** of the discrepancy-aware compression as a plug-in module but also highlights its **scalability and compatibility** with a wide range of methods based on Top-k and ATOMO, enabling integration into various existing compression schemes. **[@Reviewer_atF7 W1]**

- We **included** results with FedAvg (uncompressed baseline) in revised Table 1 to better highlight the absolute performance gap of our method under various compression ratios. **[@Reviewer_atF7 W2]**

- We **expanded** our experimental validation by testing our method on four **NLP datasets** (20News, AG News, CoLA, and MNLI) using DistilBERT. The results show that our method **consistently outperforms** the magnitude-based baseline in both accuracy and convergence speed. **[@Reviewer_atF7 W4]**

- We **added** a "time-to-accuracy" analysis in Section 4.2, showing that our method not only **improves final accuracy** but also **accelerates convergence**. **[@Reviewer_yzYJ W1]**

- We **added an analysis in Appendix A.3** to demonstrate that our discrepancy-aware method can be integrated into PowerSGD. We clarified the compatibility of our method with low-rank approximation techniques like PowerSGD, and explained why we chose ATOMO as the low-rank baseline in our experiments. **[@Reviewer_yzYJ W2&Q2]**

- We **clarified** that our focus is on compression methods and the selection rules used within them, which differ fundamentally from the core ideas of quantization. While our experiments center on compression techniques, we acknowledged that the underlying principle of our method is extensible to quantization. **This potential extension is detailed in Appendix A.4**, where we provide a comprehensive analysis and suggest it as a promising direction for future research. **[@Reviewer_yzYJ W3&Q1]**

- We have taken care to **proofread** the entire manuscript to identify and fix syntax issues, to improve the readability of the revised version. **[@Reviewer_yzYJ Q3]**

Best regards,

Authors

---

### Meta-Review · Area_Chair_s5aq · 2026-01-02

**Summary:**

**Paper Summary:**
The paper proposes a discrepancy-aware compression method for federated learning that uses local calibration data to guide compression decisions, improving accuracy and convergence under severe communication constraints.

**Strengths:**

1. The paper addresses an important problem in federated learning with a novel and intuitive idea.
2. It demonstrates strong empirical gains over magnitude-based baselines across multiple datasets, models, and compression types.
3. It provides theoretical convergence guarantees and compatibility with error-feedback mechanisms.
4. It is designed as a plug-in module, making it broadly applicable to existing compression schemes.

**Weaknesses:**

1. The Computational overhead from calibration is non-negligible; justification relies on limited scenarios.
2. The scope of baselines is narrow. It does not compare against advanced quantization-based or hybrid methods.
3. The low-rank experiments focus on ATOMO; lack of PowerSGD results weakens practical claims.
4. The initial experiments were limited to vision tasks; NLP results were added later but remain less detailed.
5. There are some wording issues and dismissive characterization of existing methods.

**Reviewer Concerns:**

Most major concerns were addressed, including convergence guarantees, EF integration, FedAvg baseline, NLP experiments, and time-to-accuracy analysis.

There remains some outstanding issues:
1. Empirical validation on PowerSGD and other advanced compression/quantization methods.
2. Broader baseline coverage to fully justify overhead.

**Reviewer Scores:**

Since most concerns have been addressed, I think some reviewers who originally gave a score of 4 may increase the score to 6.

---

### Decision · Program_Chairs · 2026-01-26

Accept (Poster)